# Evaluating APSIM-and-DSSAT-CERES-Maize Models under Rainfed Conditions Using Zambian Rainfed Maize Cultivars

**Charles B. Chisanga** [1,*] , **Elijah Phiri** [2] and **Vernon R. N. Chinene** [2]

1   School of Natural Resources, Plant and Environmental Sciences, Copperbelt University, P. O. Box 21692 Kitwe, Zambia
2   School of Agricultural Sciences, Department of Soil Science, University of Zambia, P. O. Box 32379 Lusaka, Zambia; ephiri@unza.zm (E.P.); vernon.chinene@unza.zm (V.R.N.C.)
*   Correspondence: cbchisanga@gmail.com or charles.chisanga@cbu.ac.zm; Tel.: +26-0955-885667

**Abstract:** Crop model calibration and validation is vital for establishing their credibility and ability in simulating crop growth and yield. A split–split plot design field experiment was carried out with sowing dates (SD1, SD2 and SD3); maize cultivars (ZMS606, PHB30G19 and PHB30B50) and nitrogen fertilizer rates (N1, N2 and N3) as the main plot, subplot and sub-subplot with three replicates, respectively. The experiment was carried out at Mount Makulu Central Research Station, Chilanga, Zambia in the 2016/2017 season. The study objective was to calibrate and validate APSIM-Maize and DSSAT-CERES-Maize models in simulating phenology, mLAI, soil water content, aboveground biomass and grain yield under rainfed and irrigated conditions. Days after planting to anthesis (APSIM-Maize, anthesis (DAP) RMSE = 1.91 days; DSSAT-CERES-Maize, anthesis (DAP) RMSE = 2.89 days) and maturity (APSIM-Maize, maturity (DAP) RMSE = 3.35 days; DSSAT-CERES-Maize, maturity (DAP) RMSE = 3.13 days) were adequately simulated, with RMSEn being <5%. The grain yield RMSE was 1.38 t ha$^{-1}$ (APSIM-Maize) and 0.84 t ha$^{-1}$ (DSSAT-CERES-Maize). The APSIM- and-DSSAT-CERES-Maize models accurately simulated the grain yield, grain number m$^{-2}$, soil water content (soil layers 1–8, RMSEn $\leq$ 20%), biomass and grain yield, with RMSEn $\leq$ 30% under rainfed condition. Model validation showed acceptable performances under the irrigated condition. The models can be used in identifying management options provided climate and soil physiochemical properties are available.

**Keywords:** APSIM-Maize; calibration; CERES-Maize model; grain yield; RMSE; maize; validation





## 1. Introduction

Maize (*Zea mays* L.) is the third important cereal crop after wheat and rice in the world [1]. It is grown both in the warmer temperate regions and humid subtropics [2]. In tropical Africa, nearly all maize grain is used for human food and prepared and consumed in many ways [1,2]. The generation of new research data using field experiments and the publication of such findings are not adequate to meet the increasing needs for the transference of new agro-technologies [3]. Furthermore, traditional fertilizer field trials' findings are seasonal, site-specific, laborious and costly. The experimental data for one site may not be replicated at another due to spatial variations in the soil type and climate. In contrast, most maize agronomic recommendations are based on specific sets of field trials that are rarely repeatable due to environmental and seasonal variations [4].

The literature revealed that there is a need to transfer agrotechnology that enables the solving of agricultural problems through less time-consuming and expensive system-based research approaches. The use of crop simulation models (CSMs) is an efficient complement to agronomic field trails, and these are used to characterize, develop and assess crop production systems [5]. The use of system-based experiments to transfer agro-technology to farmers, policymakers and planners is feasible by using decision support tools (DSTs), such as the Agricultural Production Systems Simulator (APSIM) [6,7] and

Decision Support Systems for Agrotechnology Transfer (DSSAT) [3,8,9]. Policymakers and land managers can use crop models to identify management options, provided climatic data, soil physiochemical properties, socioeconomic information and management are available [10]. The interactions between weather, management and soil physiochemical properties that affect crop growth and yield can be assessed using crop models [11]. APSIM and DSSAT allow users to conduct onscreen experiments in minutes on a workstation, server, laptop or desktop computer [12].

The APSIM has been used to simulate wheat evapotranspiration, yield, nitrogen uptake and soil evaporation [13]. APSIM has been evaluated in Asia under different environments, crops and management scenarios. The model performed well in simulating different cropping systems and multi-crop sequences [14]. However, the model was unable to accurately simulate aspects related to high temperatures, salinity, conservation agriculture and soil micronutrients deficiencies. APSIM has been used in predicting soil water deficits and drainage in New Zealand [15]. Additionally, the APSIM model more sensitive to management practices and environmental conditions.

The APSIM-Maize and CERES-Maize models were assessed [16]. The APSIM and DSSAT models are different in the way they model incoming solar radiation and photosynthetic active radiation (PAR). APSIM uses the Beer's law approach [17], while DSSAT uses the approach by [18]. The evaporation algorithm developed [19] is used in APSIM-Maize and CERES-Maize models. The APSIM-maize and DSSAT-CERES-Maize both require many inputs, with many model parameters not readily available at the farm level. APSIM parameterization and calibration requires obtaining better growth and development descriptions of the low-yielding cultivars found in African farming systems [20]. APSIM and DSSAT-CERES-Maize models have been used in long-term cropping systems trials, on-farm decision-making, farming systems design for production, seasonal climate forecasting, risk assessment and education in tropical and semi-arid areas [6,21,22].

The DSSAT-CERES-Maize model has been evaluated under a wide range of field practices and environmental conditions [3,8,23]. It has also been evaluated in Zimbabwe, Malawi, Kenya, South Africa and Zambia [2,4,24]. The ZMS606 had previously been calibrated for the DSSAT-CERES-Maize model under rainfed conditions in the 2013/2014 season [2]. ZMS606 is a medium (125–130 days) maturing cultivar commonly grown by small-scale farmers throughout Zambia. The DSSAT-CERES-Maize model accurately predicts the crop response to nitrogen, nitrogen uptake and crop yield variability, and it has been used to explore the potential of new cultivars for new areas and optimal sowing dates without establishing costly field trials [25].

The use of APSIM-Maize and DSSAT-CERES-Maize models is limited in Sub-Saharan Africa due to the scarcity of suitable model input data for model parameterization, calibration, testing, validation and application. The parameterization and calibration of the APSIM-Maize and DSSAT-CERES-Maize models is necessary for new cultivars, environments and subsequent model evaluations to ensure the credibility of the model performances [26]. The performances of the APSIM and DSSAT-CERES-Maize models have been tested in different locations, environments and under long-term cropping systems and climate change [6,9]. Although APSIM plant and soil modules have been extensively calibrated and evaluated under Australian and Eastern and Southern African conditions [6,22], they have not been comprehensively calibrated and tested under Zambian conditions. Therefore, the study objectives were: (i) to determine the CSPs of three maize cultivars for parameterization of the APSIM-Maize and DSSAT-CERES-Maize models under rainfed conditions; (ii) to evaluate the APSIM-Maize and DSSAT-CERES-Maize models in predicting the grain, biomass yield, leaf area index and soil water content by maize under three sowing dates (SDs) and nitrogen fertilizer rates under rainfed conditions and (iii) to validate the APSIM-Maize and DSSAT-CERES-Maize models using N fertilizer levels under irrigated conditions.

## 2. Materials and Methods

### 2.1. Study Site and Soil Data

The study was at the Mount Makulu Central Research Station in Chilanga, Zambia (latitude: 15° 550′ S, longitude: 28° 250′ E, altitude: 1213 m). The daily weather data (solar radiation, maximum and minimum temperature and rainfall) were collected from the automatic meteorological stations for the Zambia Meteorological Department (ZMD) near the experimental site. The crop models required accurate solar radiation to drive their simulation of photosynthesis and the carbon balances that govern plant growth. The soil type at Mount Makulu is classified in Soil Taxonomy as Ustic Paleustalf [27].

### 2.2. Weather Data under Rainfed Condition

The daily precipitation, solar radiation, maximum and minimum temperature data are presented in Figure 1 for the 2016/2017 season. The total precipitation, mean, maximum and minimum temperature during the field experiment period were 930.17 mm, 21.83 °C, 28.29 °C and 15.36 °C, respectively. As the season progressed, the precipitation, solar radiation, maximum and minimum temperatures reduced. SD1 (precip: 850.37 mm, tmax: 27.47 °C, tmin: 17.14 °C, tmean: 21.72 °C and Srad: 17.38 MJ m$^{-2}$ day$^{-1}$) recorded higher meteorological parameters compared to SD2 (precip: 763.27 mm, tmax: 27.02 °C, tmin: 16.32 °C, tmean: 21.67 °C and Srad: 17.22 MJ m$^{-2}$ day$^{-1}$) and SD3 (precip: 515.27 mm, tmax: 26.88 °C, tmin: 17.16 °C, tmean: 21.24 °C and Srad: 17.38 MJ m$^{-2}$ day$^{-1}$). All the three SDs experienced mean maximum temperatures below 38 °C during the growing period from emergence to physiological maturity. The critical temperature that can have detrimental effects on the maize yield is 32 °C, as reported [28]. Maize during its growing period requires about 450–600 mm of water season$^{-1}$, which is taken from the soil moisture reserves. SD3 received little water during the season, and this most likely affected the plant growth and yield.

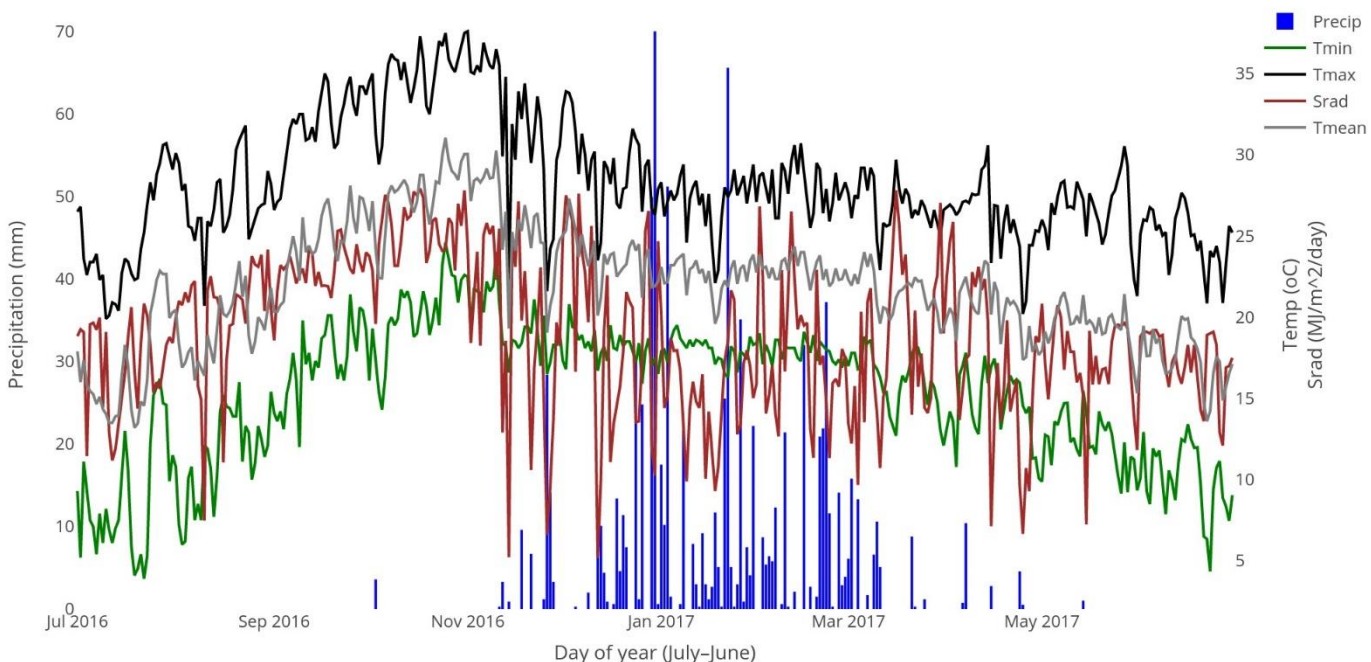

**Figure 1.** Daily weather data collected during the rainfed field experiment at Mt Makulu during the 2016/2017 season.

### 2.3. Weather Data under Irrigation Condition

The weather data (rainfall, maximum, minimum temperature and solar radiation) used in the validation was obtained from the Zambia Meteorological Department (ZMD), as shown in Figure 2. The weather data was taken from May 2016 to November 2016. The Tmin, Tmax, Tmean and precipitation were 13.76 °C, 28.64 °C, 21.21 °C and 71.70 mm for

the whole period, respectively. The mean temperature ranged from 13.20 °C to 31.05 °C under the irrigated conditions. It has been observed that maize grows well in between mean temperatures of 15 and 35 °C [29].

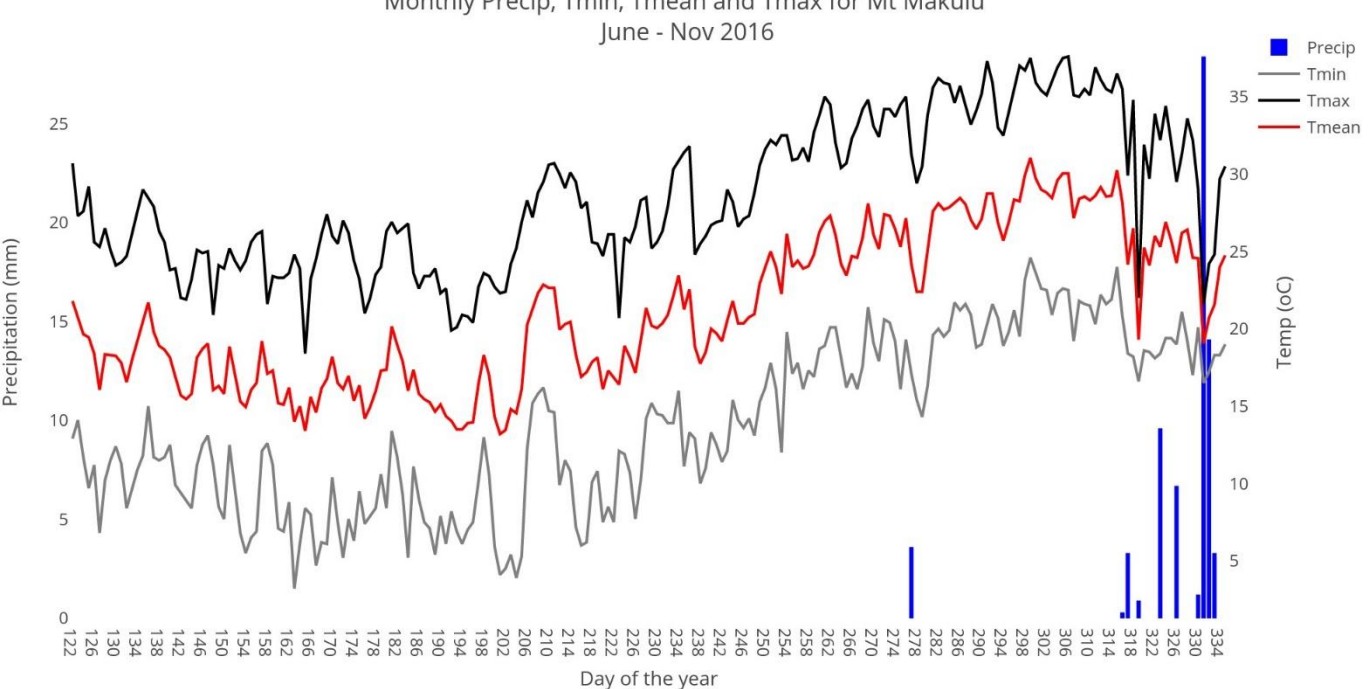

**Figure 2.** Daily weather data (June–November 2016) collected during the irrigated field experiment at Mount Makulu.

*2.4. Experimental Design, Treatments, Soil Physical and Chemical Analysis*

A rainfed split–split plot design was set up at the Mount Makulu Central Research Station in Chilanga, Zambia with three sowing dates (SDs); three maize cultivars (ZMS 606, PHB 30G19 and PHB 30B50); three nitrogen (67.20, 134.40 and 201.60 kg N ha$^{-1}$) fertilizer rates and three replicates, respectively. The main plot was the SD (3 sowing dates [SD]) at 14-day intervals, while the subplots and sub-subplots were the maize cultivars and N rates. Individual plot sizes were 6 m (7 rows) by 5 m (30 m$^2$). The plots were separated from each other by a 2-m distance to prevent the cross-contamination of treatments. Three seeds were sown by hand at a 5-cm depth in a flat seedbed in 0.75-m row spacing and 0.50-m spacing between plants per station and later thinned to two plants. The initial soil conditions were sampled using a soil auger at 20-cm intervals until a 100-cm depth, weighed and oven-dried at 105 °C.

The soil parameters presented in Table 1 were analyzed at the Mount Makulu Central Research Station in Chilanga, Zambia. The soil pH in calcium chloride ranged from 6.2 to 6.80, considered to be within the neutral to optimal range for crop growth. The pH was considered to be favorable for availability of most plant nutrients. The total nitrogen percentage values ranged from 0.031 to 0.061 and, therefore, were considered to be very low. The soil organic carbon percent (OC) was analyzed using the Walkley and Black method [30], and the values ranged from 0.35 to 0.80 and were considered to be low. The critical OC value is 1.58%. The phosphorus (P) and potassium (K) ranged from 10 to 18 mg kg$^{-1}$ and 0.59 to 1.12 mg kg$^{-1}$, respectively. The P ranged from very low to low, while the K was very low. The soil texture and organic matter were used as inputs into the Soil–Plant–Air–Water (SPAW) pedo-transfer functions [31,32] to determine the lower limit (LL), drained upper limit (DUL), saturation (SAT), hydraulic conductivity and bulk density (Table 1). Higher organic matter increased the water-holding capacity and conductivity, as a result of pore space distribution and soil aggregation [32]. In addition, the organic matter and soil texture are two variables that affect the soil water content.

**Table 1.** Physical and chemical soil analyses under rainfed conditions. Reproduced with permission from [33].

| Depth (Cm) | 0–20 | 20–40 | 40–60 | 60–80 | 80–100 | Analysis Method |
|---|---|---|---|---|---|---|
| pH (water) | 7.30 | 7.20 | 7.50 | 7.70 | 7.60 | 1:5 soil water |
| Total N (%) | 0.031 | 0.042 | 0.054 | 0.061 | 0.036 | Modified Kjeldahl method |
| $NO_3N$ | 29.90 | 48.70 | 56.40 | 70.10 | 42.80 | |
| $NH_4N$ | 18.00 | 29.20 | 33.90 | 42.10 | 25.70 | |
| P extractable (mg $kg^{-1}$) | 10.00 | 11.00 | 10.00 | 18.00 | 12.00 | Bray 1 |
| K (mg $kg^1$) | 1.05 | 0.99 | 1.12 | 0.59 | 0.89 | Ammonium acetate |
| Ca (cmol(+) $kg^{-1}$) | 11.00 | 9.30 | 3.40 | 2.90 | 3.20 | Ammonium acetate |
| Mg (cmol(+) $kg^{-1}$) | 3.50 | 2.70 | 2.30 | 1.00 | 1.30 | Ammonium acetate |
| OC (%) | 0.35 | 0.57 | 0.66 | 0.82 | 0.50 | Walkley & Black method |
| OM (%) | 0.602 | 0.980 | 1.135 | 1.410 | 0.860 | |
| CEC (cmol(+) $kg^{-1}$) | 15.57 | 13.02 | 6.85 | 4.52 | 5.42 | Ammonium acetate |
| Bulk density (g $cm^{-3}$) | 1.43 | 1.41 | 1.41 | 1.46 | 1.36 | SPAW |
| Silt (%) | 12.80 | 16.80 | 12.80 | 18.80 | 2.80 | Hydrometer method |
| Sand (%) | 39.60 | 35.60 | 37.60 | 41.60 | 37.60 | |
| Clay (%) | 47.60 | 47.60 | 49.60 | 39.60 | 59.60 | |
| Soil texture | clay | clay | clay | clay | clay | SPAW |
| LL | 0.287 | 0.287 | 0.299 | 0.244 | 0.350 | SPAW |
| DUL | 0.407 | 0.409 | 0.419 | 0.363 | 0.470 | |
| SAT | 0.459 | 0.467 | 0.468 | 0.447 | 0.487 | |
| SHC (mm $h^{-1}$) | 0.350 | 0.500 | 0.290 | 1.480 | 0.010 | |

Phosphorus; LL: lower limit (Wilting point); DUL: drained upper limit (Field Capacity); SAT: saturation; SHC: Saturated hydraulic conductivity; SPAW: Soil–Plant–Air–Water.

### 2.5. Soil Water Content Measurement

The Diviner 2000 series II probe (Senteck Sensor Technologies, Stepney, SA, Australia; https://sentektechnologies.com/product-range/soil-data-probes/diviner-2000/) is a portable and robust device measuring soil water at multiple depths (at 10-cm intervals) in the profile [34]. The Diviner 2000 series II probe consists of a probe and handheld data logging display unit. It measures the relative changes in the volumetric soil moisture content, which is used to show the most important soil water trends with time. Access tubes were installed in each experimental plot and soil water content measurement, taken two to three times per week using a Diviner 2000 series II probe. The series of readings taken by the Diviner 2000 series II assisted in showing trends of crop water use in the soil profile. The volumetric soil water content collected using the Diviner 200 series II probe were used as inputs and simulated with the APSIM-Maize and DSSAT-CERES-Maize models.

### 2.6. Plant Materials

The maize cultivars PHB 30G19, PHB 30B50 and ZMS 606 have medium maturity with the comparative relative maturity of 120–130 days. These cultivars were selected as the major cultivars planted by small-scale farmers and their long commercial lives. PHB30B50 is recommended to be grown under irrigation, while PHB30G19 and ZMS 606 can be grown under irrigated and rainfed conditions in Zambia. PHB30G19 (white) and PHB30B50 (yellow) are produced by DuPont Pioneer (Lusaka, Zambia). ZMS 606 is an exceptionally good drought-tolerant maize cultivar produced by ZamSeed (Lusaka, Zambia). The selected cultivars can be grown in all the three agro-ecological regions of Zambia.

### 2.7. Plant Growth Analysis

A plant growth analysis was observed at the vegetative (emergence, V6) and reproductive (silking (R1), dough stage (R4) and physiological maturity (R6)) stages and recorded when 50% and 75% of the plants reached the stages, respectively, as described in [35,36] (Tables 2 and A1). The biomass from all the subplots was harvested at the recommended growth stages. The dry weight of each fraction of biomass harvested at each vegetative and reproductive stage was recorded before and after drying the samples at 70 °C for three days.

Four plants were sampled from each subplot at V6, R1 and R4 for the biomass and leaf area measurements. The maize leaf area measurement was calculated by multiplying the manually measured length and maximum width by 0.75, reported as the maize calibration factor [37]. Accurate prediction of the leaf area measurement is critical for the precise forecasting of crop growth and yield by most crop simulation models [9,36]. The final maize aboveground biomass was harvested using standard procedures [36].

**Table 2.** Growth and development stages of maize. Reproduced with permission from [38].

| Vegetative Stage | Reproductive Stage |
| --- | --- |
| Emergence (VE) | silking (R1) |
| first leaf with collar (V1) | blister (R2) |
| second leaf with full collar (V2) | milk (R3) |
| third leaf with full collar (V3) | dough (R4) |
| nth leaf with full collar (V(n)) | dent (R5) |
| tasseling (VT) | maturity (R6) |

*2.8. Modelling Approach*

2.8.1. Description of the APSIM-Maize Module

APSIM version 7.9 [21] used in this study includes the APSIM-Maize, SoilWater, SoilN and SurfaceOM modules. It is a process-based model that runs at a daily time step and combines biophysical and management modules within a central engine to simulate the crop growth, development, soil water, phosphorus, carbon and nitrogen dynamics using the current and future climate scenarios [22,26,39]. It contains more than 80 modules that simulate plant, animal, soil, climate and management scenarios. It can simulate more than 20 crops and forests, such as alfalfa, eucalyptus, cowpea, pigeon pea, peanuts, cotton, lupin, maize, wheat, barley, sunflower, sugarcane, chickpea and tomato [7].

The APSIM-Maize module [40] consists of a phenology model, structure model and an arbitrator for nitrogen and biomass partitioning to various plant organs in the model. It simulates maize growth, leaf area and development on a daily time step. The growth of maize responds to precipitation, temperature, radiation, soil nitrogen (SoilN) and soil water supply (SoilWat) [41,42]. The phenology model simulates the growth stages, and each stage is divided into subphases. The duration of each phase is regulated by the temperature and photoperiod and may be affected by water and nitrogen (N) stresses.

2.8.2. Description of the DSSAT-CERES-Maize Model

The Decision Support System for Agrotechnology Transfer (DSSAT) v4.7 comprises 42 crop simulation models (CSMs), and this model is the CERES-Maize [36,43,44]. The DSSAT-CERES (Crop Environmental Resource Synthesis) maize model [45,46] is a process-oriented, management level model that simulates soil water and nitrogen dynamics, crop growth, yield and management scenarios at the field level on a daily time step from inputs of climate data, genotype, soil and management [3,45,47]. The soil water balance in DSSAT-CERES-Maize is based on Ritchie's model, where the concept of a drained upper limit (DUL) and drained lower limit (LL) of the soil is used as the basis of the available soil water [48,49]. It is a widely used maize model and a reference for comparing new advances in maize simulations by farmers, researchers and planners in many countries [9].

2.8.3. Input Dataset into the APSIM-Maize and DSSAT-CERES-Maize Models

The minimum input data required by the APSIM-Maize and DSSAT-CERES-Maize models are: the minimum input data required to run APSIM-Maize and DSSAT CERES-Maize models includes daily records of rainfall, solar radiation, maximum and minimum air temperature, soil characterization data (physical and chemical properties and morphological properties for each layer), soil drained upper limit, soil lower limit, soil saturation and genetic cultivar coefficients characterizing the cultivar being grown (plant development and grain biomass and crop management information). The soil water balance is

simulated to evaluate the potential yield reduction caused by soil water deficits. The data presented in Tables 1 and 3 were used as inputs into the crop simulation models.

**Table 3.** Treatment effect of the sowing date, cultivar and N on the grain wt., grain number m$^{-2}$, grain wt., stover wt., aboveground biomass wt. and cob wt. under rainfed conditions.

| Treatment/ Cultivar | Grain wt. (g m$^{-2}$) | Grain No m$^{-2}$ | Grain Unit wt. (g) | Stover wt. (g m$^{-2}$) | Biomass wt. (g m$^{-2}$) | Cob wt. (g m$^{-2}$) |
|---|---|---|---|---|---|---|
| SD1 | 907.6 [a] | 2153 [a] | 42.35 [a] | 277.5 [a] | 1185.0 [a] | 158.0 [a] |
| SD2 | 716.3 [b] | 2064 [a] | 34.27 [b] | 258.0 [b] | 974.3 [b] | 140.8 [ab] |
| SD3 | 617.7 [c] | 2014 [a] | 31.86 [b] | 314.5 [b] | 932.2 [b] | 128.4 [b] |
| Significance | *** | ns | *** | *** | *** | *** |
| Tukey HSD 5% | 90.78 | 17.90 | 1.25 | 31.30 | 115.26 | 24.95 |
| CV % | 18.52 | 57.8 | 4.6 | 16.84 | 17.05 | 5.1 |
| ZMS 606 | 732.6 [a] | 2258 [a] | 33.62 [b] | 234.5 [b] | 967.1 [a] | 109.10 [b] |
| P30B19 | 733.9 [a] | 2210 [a] | 34.52 [b] | 318.0 [a] | 1052.0 [a] | 157.8 [a] |
| P30G50 | 775.2 [a] | 1763 [b] | 40.34 [a] | 297.4 [a] | 1073.0 [a] | 160.3 [a] |
| Significance | ns | *** | *** | *** | ns | *** |
| Tukey HSD 5% | 61.78 | 211.38 | 2.803.35 | 24.20 | 80.01 | 11.43 |
| CV % | 15 | 15.52 | 14.12 | 15.5 | 14.1 | 3.3 |
| Nitrogen (N) rate (N1) | 742.3 [a] | 2061 [a] | 36.70 [a] | 278.2 [a] | 1020.0 [a] | 137.37 [a] |
| Nitrogen (N) rate (N2) | 729.2 [a] | 2054 [a] | 35.91 [a] | 272.6 [a] | 1002.0 [a] | 137.74 [a] |
| Nitrogen (N) rate (N3) | 770.2 [a] | 2116 [a] | 35.87 [a] | 299.2 [a] | 1069.0 [a] | 152.07 [a] |
| Significance | ns | ns | ns | ns | ns | ns |
| Tukey HSD 5% | 90.79 | 13.74 | 3.43 | 31.88 | 124.6 | 16.18 |
| CV % | 18.5 | 44.2 | 16 | 18.52 | 20.4 | 5.6 |
| Interaction (SD * V) significance | ns | ns | ** | ** | ns | ns |
| Interaction (V * N) significance | ns | ns | ns | ns | ns | ns |
| Interaction (V * SD * N) significance | ns | ns | ns | ns | ns | ns |

Means sharing the same letter in the table do not differ statistically at $p < 0.05$; LSD = Least Mean Differences; ** = highly significant at the 5% level; *** = very highly significant at the 5% level; ns = Nonsignificant; At R6, weight = g m$^{-2}$ square meter (g m$^{-2}$ * 10 = t ha$^{-1}$); wt. = weight.

### 2.9. Parameterization, Calibration and Validation of the APSIM-Maize and DSSAT-CERES-Maize Models

#### 2.9.1. Parameterization and Calibration of the APSIM-Maize and DSSAT-CERES-Maize Models

The soil analysis and aboveground biomass data (Tables 1 and 3) collected from the 2016/2017 rainfed field experiment was used as inputs to calibrate the APSIM-Maize and DSSAT-CERES-Maize models. In APSIM, U and CONA parameters similar to the DSSAT-CERES-Maize model are used to determine the first and second evaporation stages. The U and CONA were set at 6 mm and 3.5 mm day$^{-1}$, respectively and these parameters are crop-specific and determine the rate of root extension and lower limit of crop water extraction. The Air-dry (mm mm$^{-1}$) soil moisture limit at which soil dries by evaporation was calculated as 0.5 × LL15 (015 cm soil layer), 0.9 × LL15 (1530 cm soil layer) and at deeper depths same as LL15 [50]. The crop lower limit (CLL in mm mm$^{-1}$) was taken as equal to LL15. The soil water characteristics used as model inputs are shown in Table 1 above.

The growth and development module of APSIM-Maize uses a set of different coefficients as presented in Table 3 to define the phenology, crop growth and yield [51,52]. In the APSIM-Maize model, the B_130 cultivar was used as the starting point for calibrating the ZMS606, PHB 30G19 and PHB 30B50 maize cultivars. The Cultivar Specific Parameters

(CSPs) for the three maize cultivars were obtained step-by-step based on the observed phenological stages (days after planting (DAP) to anthesis and maturity; leaf area index (LAI); biomass and grain yield (grain growth rate, grain yield, grain unit wt. and grain number)) using the manual trial and error method until the simulated values were within 9–20% of the observed values [53].

The CSPs in the DSSAT-CERES-Maize model were calibrated automatically using the GLUE (Generalized Likelihood Uncertainty Estimation) program embedded within DSSAT [43]. The CSPs describe the phenology and grain yield. Six parameters (P1, P2, P5, G2, G3 and PHINT), as shown in Table 2, are used by the DSSAT-CERES-Maize model [3,23]. The final leaf number in DSSAT-CERES-Maize is calibrated using the PHINT CSP. Using the soil and weather data, calibration procedure was done for the phenology stages, followed by grain development. The "P" CSPs (P1, P2 and P5) and "G" CSPs (G2 and G3) were computed using the phenological dates and grain yield, respectively. The derived CSPs for the cultivars used in the field experiment were added to the DSSAT's genotype file to be used in the simulations. They control the growth and development of the crop and, by affecting the root growth and root water uptake, indirectly impact the soil water content [54]. The APSIM-Maize and DSSAT-CERES-Maize models were calibrated using phenology, physiological maturity, maximum leaf area index (mLAI), grain yield, aboveground biomass yield, grain unit wt., grain number m$^{-2}$ and root soil water content to the depth of one meter.

### 2.9.2. Validation of the APSIM-Maize and DSSAT-CERES-Maize Models

The APSIM-Maize and DSSAT-CERES-Maize models were validated using the phenology, physiological maturity, mLAI, grain yield, aboveground biomass yield, grain unit wt. and grain number m$^{-2}$. Cultivar specific parameters (CSPs) derived from the calibration of the two crop models were used to evaluate the robustness of the APSIM-Maize and DSSAT-CERES-Maize models in a subtropical environment of Zambia. The observed yield and yield components from the irrigated field experiment in 2016 were used to validate the two crop models. The validation of the APSIM-Maize and DSSAT-CERES-Maize models was essential for model application.

### 2.10. Data Analysis and Model Evaluation
Analysis of Variance

The treatment effects of the sowing date (SD) on the maize cultivar and N rate on the maize yield and yield components collected from the rainfed field experiment was computed using the analysis of variance for the split–split plot design. The analysis of variance was calculated using the *ssp.plot* function in the R programming software Agricolae package and means separated at $p \leq 5$ using Tukey's HSD test [55]. On the other hand, the treatment effect of the nitrogen fertilizer rate and cultivar on the maize yield and yield components observed at the irrigated experimental site was evaluated by a split–plot analysis of variance using the *spp.plot* function in the Agricolae package [55] in R Programming software.

### 2.11. Evaluation of the APSIM-Maize and DSSAT-CERES-Maize Models

The root mean square error (RMSE), normalized (RMSEn) root mean square error, d-stat (d) and 1:1 graphs (observed vs. simulated graphs) [45,56,57] were used to evaluate the crop models. In an adequately calibrated crop model, the EF should be above 0.70 [41]. The d-stat (d) ranges from 0 (the simulated is similar to the observed mean) to 1 (perfect model performance) [58]. The simulation is considered excellent with RMSEn <10%, good if 10–20%, acceptable or fair if 20–30% and poor >30% [59]. The d $\geq$ 0.70 is considered acceptable in crop modeling and evaluations [60].

$$\text{RMSE} = [\text{N}^{-1} \sum_{i=1}^{n} (\text{P}_i - \text{O}_i)^2]^{0.5} \ \text{N}^{-1} \tag{1}$$

$$\text{RMSEn} = \frac{\text{RMSE}}{\overline{O}} * 100\% \tag{2}$$

$$\text{d} - \text{stat} = 1 - \left[ \frac{\sum (P_i - O_i)^2}{\sum (|P_i'| - |O_i'|)^2} \right] \tag{3}$$

$$\text{EF} = 1 - \left( \frac{\sum (P_i - O_i)^2}{\sum (O_i - \overline{O})^2} \right) \tag{4}$$

where: $P_i$, $O_i$, $\overline{O}$ and n are the simulated, observed, average observed values and the number of observations for the studied variables, respectively.

## 3. Results and Discussions

### 3.1. Analysis of Variance for the Sowing Dates, Cultivars, N Rate and Yield Components

3.1.1. Rainfed Condition

The treatment effect of the sowing date (SD), cultivar and nitrogen fertilizer application rates on the yield and yield components of maize are shown in Tables 3 and 4. The treatment effect of SD and the cultivar were significant on the 100-grain wt., stover wt., cob wt., stem, wt., leaf blade wt., cob width and harvest index, as reported by [61,62]. The differences in the 100-grain wt. may have been caused by differences in the initial sizes of the spikelets, in growth rates during the exponential and linear phases of the grain-filling period [61]. The soil moisture stress after silking notably decreased the pooled values for the 100-seed weight (g) with an increasing N rate. The grain yield, aboveground biomass, grain number m$^{-2}$, cob wt., cob width, harvest index and 100-grain wt. were reduced with a delay of 7 and 14 days in sowing, low soil fertility, lowering of the air temperature and reduction in cumulative rainfall during the duration of plant growth, as shown in Tables 3 and 4. The cultivar treatment effect on the grain number m$^{-2}$, 100-grain wt., stover wt., cob wt., husk wt., stem wt., leaf blade wt., harvest index and cob width was statistically significant. The plant LAI at any growth stage indicates the size of the assimilatory system, which contributes to dry matter accumulation and partitioning [63]. There was an interaction effect between the SD and cultivar on 100-grain wt. Furthermore, there was an interaction effect between the cultivar and nitrogen application rates on the stem wt. The nitrogen treatment effect on the yield and yield components was statistically nonsignificant. The application of N fertilizer did not significantly improve the maize yield [64]. However, deficiency or excess nitrogen application can reduce the maize yield [65].

The reduction in grain yield from SD1 to SD2 (1.91 t ha$^{-1}$), SD1-SD3 (2.90 t ha$^{-1}$) and SD2 to SD3 (0.99 t ha$^{-1}$) were 21.04%, 31.83% and 13.83%, respectively. A reduction in the maize yield with a delay in SD was observed in Northern Nigeria [66]. Delayed SD reduces the duration of the grain filling process and yield [24]. The SD is a critical factor in capturing sufficient solar radiation with adequate soil moisture and fertility. The sowing date considerably affected the maize grain and biomass yield [2].

**Table 4.** Treatment effect of the sowing date, cultivar and N on the husk wt., stem wt., leaf blade wt., harvest index, cob width and cob length under rainfed conditions.

| Treatment/Cultivar | Husk wt. (g m$^{-2}$) | Stem wt. (g m$^{-2}$) | Leaf Blade wt. (g m$^{-2}$) | HI | Cob Width (cm) | Cob Length (cm) |
|---|---|---|---|---|---|---|
| SD1 | 33.30 [b] | 47.77 [b] | 20.78 [b] | 0.77 [a] | 5.49 [a] | 21.43 [a] |
| SD2 | 28.06 [b] | 47.59 [b] | 25.20 [b] | 0.73 [b] | 5.29 [b] | 21.3 [a] |
| SD3 | 55.06 [a] | 68.17 [a] | 35.03 [a] | 0.66 [c] | 5.04 [c] | 18.43 [b] |
| Significance | * | *** | *** | *** | *** | *** |
| Tukey HSD 5% | 13.40 | 39.78 | 6.45 | 0.02 | 0.17 | 1.94 |
| CV % | 45.70 | 12.62 | 31.60 | 4.64 | 5.10 | 11.41 |
| ZMS 606 | 37.73 [ab] | 51.23 [b] | 19.89 [b] | 0.75 [a] | 5.17 [b] | 19.48 [a] |
| P30B19 | 44.72 [a] | 62.15 [a] | 31.59 [a] | 0.69 [c] | 5.51 [a] | 20.78 [a] |
| P30G50 | 33.97 [b] | 50.15 [b] | 29.53 [a] | 0.72 [b] | 5.14 [b] | 20.91 [a] |
| Significance | * | ** | *** | *** | *** | ns* |
| Tukey HSD 5% | 2.18 | 31.56 | 5.77 | 0.02 | 0.16 | 1.85 |
| CV % | 32.00 | 13.72 | 36.00 | 3.8 | 3.30 | 7.8 |
| Nitrogen (N) rate (N1) | 21.83 [a] | 54.88 [a] | 271.2 [a] | 0.72 [a] | 5.27 [a] | 20.33 [a] |
| Nitrogen (N) rate (N2) | 19.53 [a] | 51.56 [a] | 279.1 [a] | 0.73 [a] | 5.24 [a] | 20.33 [a] |
| Nitrogen (N) rate (N3) | 20.49 [a] | 57.08 [a] | 310.3 [a] | 0.72 [a] | 5.31 [a] | 20.51 [a] |
| Significance | ns | ns | ns | ns | ns | ns |
| Tukey HSD 5% | 2.34 | 37.93 | 54.34 | 0.02 | 0.10 | 0.94 |
| CV % | 26.20 | 15.32 | 27.60 | 4.1 | 5.60 | 16.5 |
| Interaction (SD ∗ V) significance | ns | ns | ns | ns | ns | *** |
| Interaction (V ∗ N) significance | ns | * | ns | ns | ns | ns |
| Interaction (V ∗ SD ∗ N) significance | ns | ns | ns | ns | ns | ns |

Means sharing the same letter in the table do not differ statistically at *p* < 0.05; LSD = Least Mean Differences; * = significant at the 5% level; ** = highly significant at the 5% level; *** = very highly significant at the 5% level; ns = Nonsignificant; At R6, weight = g m$^{-2}$ square meter (g m$^{-2}$ * 10 = t ha$^{-1}$); wt. = weight.

### 3.1.2. Irrigated Condition

The treatment effect of the cultivar and nitrogen fertilizer application rates on the yield and yield components of maize are shown in Table 5. The cultivar treatment effect under the irrigated condition significantly affected the 100-grain wt. and husk wt., as examined other authors [62]. The cultivar treatment effect on the 100-grain wt. and husk wt. was statistically significant. The treatment effect of the nitrogen fertilizer rate increased the grain wt., biomass wt., cob wt. and grain number m$^{-2}$. The grain number m$^{-2}$ increased with the higher N fertilizer application rates [67]. The highest husk yield was observed from ZMS 606 (383.94 kg ha$^{-1}$), followed by PHB 30G19 (301.56 kg ha$^{-1}$) and PHB 30B50 (267.27 kg ha$^{-1}$). Pooled data for the husk yield showed that N1 (375.48 kg ha$^{-1}$) had the highest, followed by N2 (289.70 kg ha$^{-1}$) and N3 (287.59 kg ha$^{-1}$), respectively. The grain number cob$^{-1}$ and grain number m$^{-2}$ were highly significantly affected by the N fertilizer treatment effect. The grain number m$^{-2}$ increased with the higher N fertilizer rate. The maximum and a minimum numbers of grains per ear were recorded at N3 and N2, respectively. ZMS 606 recorded the highest mean grain number of 349.8, followed by PHB 30G19 (292.6) and PHB 30B50 (281.9). The highest grain number per ear was observed at the highest nitrogen application rate [68]. Similar results were obtained for maize yield response to the N fertilizer application [64]. However, nitrogen fertilizer application rate had no effect on the HI [69].

**Table 5.** Treatment effect of the cultivar and N on the grain wt., grain number m$^{-2}$, 100-grain wt., stover wt., aboveground biomass wt., grain No. m$^{-2}$, cob wt., ear wt., husk wt., stem wt. and leaf blade wt. under irrigated conditions.

| Treatment/ Cultivar | Grain wt.(g m$^{-2}$) | Grains No. m$^{-2}$ | 100-Grain wt.(g m$^{-2}$) | Stover wt. (g m$^{-2}$) | Biomass(g m$^{-2}$) | Cob wt.(g m$^{-2}$) | Husk wt.(g m$^{-2}$) | Stem wt.(g m$^{-2}$) | HI | Leaf Blade wt.(g m$^{-2}$) |
|---|---|---|---|---|---|---|---|---|---|---|
| ZMS 606 | 717.16 [a] | 2798.67 [a] | 27.53 [c] | 266.62 [a] | 983.98 [a] | 117.21 | 38.39 | 47.68 | 0.72 [a] | 84.79 |
| PHB 30B19 | 611.70 [b] | 2341.10 [b] | 29.33 [b] | 259.05 [a] | 870.75 | 122.49 | 30.16 | 43.00 [a] | 0.80 [a] | 80.84 |
| PHB 30B50 | 662.00 [b] | 2254.94 [b] | 31.39 [a] | 272.39 [a] | 934.40 | 126.41 | 26.73 | 49.85 | 0.71 [a] | 91.54 |
| Significance | ns | ns | ** | ns | ns | ns | * | ns | ns | ns |
| LSD 5% | 104.39 | 341.36 | 1.72 | 42.191 | 127.79 | 17.28 | 10.82 | 12.77 | 0.04 | 23.53 |
| CV % | 15.31 | 23.00 | 4.47 | 18.70 | 21.8 | 24.3 | 14.50 | 19.00 | 2.70 | 23.2 |
| N rate | | | | | | | | | | |
| N1 | 565.69 [b] | 2132.55 [b] | 29.04 [a] | 250.98 [a] | 816.67 | 105.06 | 37.55 | 44.00 | 0.69 [a] | 81.53 |
| N2 | 711.47 [a] | 2562.81 [a] | 30.26 [a] | 274.09 [a] | 985.55 | 129.03 | 28.97 | 49.8 | 0.72 [a] | 89.75 |
| N3 | 713.71 [a] | 2699.35 [a] | 28.96 [a] | 273.20 [a] | 986.91 | 132.03 [a] | 28.76 | 46.73 [a] | 0.72 [a] | 85.91 |
| Significance | ** | ** | ns | ns | * | * | ns | ns | ns | ns |
| LSD 5% | 104.39 | 341.36 | 1.72 | 42.19 | 127.79 | 17.73 | 10.82 | 12.97 | 0.04 | 23.53 |
| CV % | 15.30 | 13.50 | 4.47 | 15.44 | 13.40 | 14.14 | 33.17 | 27.00 | 5.12 | 26.73 |
| Interaction (V * N) | | | | | | | | | | |
| Significance | ns | ns | ns | ns | ns | ns | ns | ns | ns | ns |

Means sharing the same letter in the table do not differ statistically at *p* < 0.05; LSD = Least Mean Differences; * = significant at the 5% level; ** = highly significant at 5%; NS = Nonsignificant; sqm = square meter; wt. = weight.

Stress as a result of water deficiency from the silking to maturity stages affected the ultimate sizes and yields of ears, and similar results were reported [28]. Adverse conditions such as water stress and nitrogen deficiency delay plant growth and slow silk development [70,71]. Nitrogen is a yield-limiting nutrient, and its application plays a significant role in improving the soil fertility [71].

### 3.2. Performance of APSIM-Maize and DSSAT-CERES-Maize Models in Simulating Growth and Yield for Three Maize Cultivars

#### 3.2.1. Phenology

The comparison between the simulated days after planting (DAP) and observed days after planting (DAP) to anthesis, physiological maturity (DAP), maximum leaf area index (mLAI), grain unit wt. at maturity, grain number m$^{-2}$, biomass and grain yield statistics are presented in Figure 3a–l and Tables 6, 7, A2 and A3 for the APSIM-Maize and DSSAT-CERES-Maize models. The percent prediction (%PD) values for anthesis and maturity at SD1 (DSSAT-CERES-Maize model) were close to or equal to zero, and this indicted excellent agreement between the observed and simulated values. The %PD for anthesis and maturity was <20% of the observed values, and similar results were reported by [53]. The DSSAT-CERES-Maize model simulated maize cultivar anthesis, and physiological maturity was 62–68 and 122–134 DAP, respectively during the 2016/2017 season. The APSIM-Maize model simulated maize cultivar anthesis, and maturity was from 64 to 67 and 123 to 130 DAP, respectively. However, the observed maturity for the maize cultivars ranged from 120 to 130 DAP. Another study indicated that delayed sowing reduced the days after planting to anthesis and the number of leaves [24].

The results showed that there was good agreement between the observed and simulated anthesis (d = 0.47–0.49) and maturity (d = 0.71–0.80) for the cultivars using both models. The modeling of anthesis and maturity for the cultivars (ZMS 606, PHB 30G19 and PHB 30B50) was excellent (RMSEn ≤ 10). In the DSSAT-CERES-Maize model, the RMSE between the observed and simulated anthesis and maturity were 2.89 (d-stat = 0.47) and 3.13 (d-stat = 0.80) days, respectively. The anthesis and maturity RMSEn were 4.36 and 2.50 and considered excellent across the three cultivars. In APSIM-Maize, the RMSE for the simulation of DAP to anthesis and maturity were 1.91d (RMSEn = 2.89%, d-stat = 0.49) and 3.35d (RMSEn = 2.68%, d-stat = 0.71) days across the cultivars, respectively.

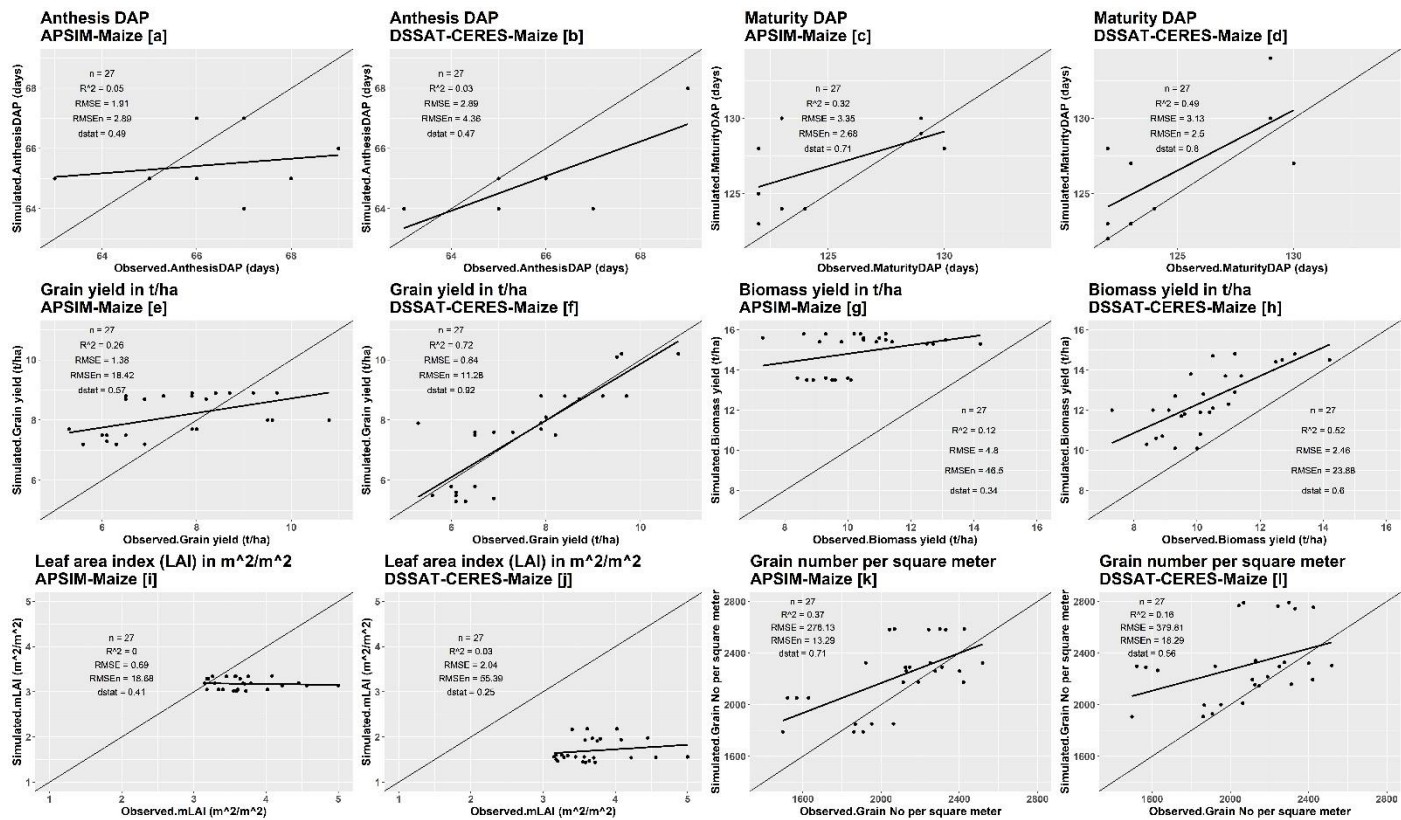

**Figure 3.** Overall calibration statistics between the observed and simulated values for the APSIM-Maize and DSSAT-CERES-Maize models using 1:1 graphs under rainfed conditions. The graphs show simulated vs. observed values for the anthesis (DAP), maturity (DAP), grain yield (t ha$^{-1}$), aboveground biomass (t ha$^{-1}$), mLAI (m$^2$ m$^{-2}$) and grain number m$^{-2}$.

**Table 6.** APSIM-Maize and DSSAT-CERES-Maize model calibration statistics for the days after planting (DAP) to anthesis for maize cultivars under rainfed conditions.

| | APSIM-Maize | | | | | DSSAT-CERES-Maize | | | | |
|---|---|---|---|---|---|---|---|---|---|---|
| Cultivar | RMSEn | d-Stat | RMSE | ME | R$^2$ | RMSEn | d-Stat | RMSE | ME | R$^2$ |
| ZMS 606 | 0.87 | 0.89 | 0.58 | −0.33 | 0.75 | 2.90 | 0.42 | 1.91 | −1.67 | 0.00 |
| PHB 30G19 | 2.74 | 0.53 | 1.83 | −0.67 | 0.06 | 1.22 | 0.93 | 0.82 | −0.67 | 0.94 |
| PHB 30B50 | 4.10 | 0.35 | 2.71 | −1.33 | 0.11 | 6.89 | 0.28 | 4.55 | −3.33 | 0.96 |

**Table 7.** The APSIM-Maize and DSSAT-CERES-Maize model calibration statistics for the days after planting (DAP) to maturity for maize cultivars under rainfed conditions.

| | APSIM-Maize | | | | | DSSAT-CERES-Maize | | | | |
|---|---|---|---|---|---|---|---|---|---|---|
| Cultivar | RMSEn | d-Stat | RMSE | ME | R$^2$ | RMSEn | d-Stat | RMSE | ME | R$^2$ |
| ZMS 606 | 3.30 | 0.63 | 4.12 | 3.00 | 0.25 | 2.32 | 0.73 | 2.89 | 2.33 | 0.82 |
| PHB 30G19 | 2.77 | 0.65 | 3.46 | 2.00 | 0.18 | 3.61 | 0.71 | 4.51 | 3.67 | 0.60 |
| PHB 30B50 | 1.73 | 0.86 | 2.16 | 0.67 | 0.84 | 1.46 | 0.91 | 1.83 | −0.67 | 0.96 |

A multi-model ensemble of the APSIM, DSSAT-CERES and Aquacrop models have been used to simulate maize growth and yield under rainfed condition in Ethiopia [72]. The APSIM-Maize and DSSAT-CERES-Maize models simulated accurately days after planting to anthesis (RMSE = 1.73–4.09) and maturity (RMSE = 1.66–5.36). It was observed by [72]

that the DSSAT-CERES-Maize model overestimated DAP to maturity for late-maturing cultivars. The results reported are comparable to this study (Tables 6 and 7) [72]. APSIM-maize has been used to simulate the optimal sowing windows for maize [73]. The model simulated accurately anthesis (DAP) and maturity (DAP), grain yield and biomass for maize cultivars.

### 3.2.2. Biomass and Grain Yields

The grain yield results showed a wide range of %PD for simulated from observed values ($-25.93$ to $+45.28\%$ for APSIM-Maize and $-21.74$ to $+49.06\%$ for DSSAT-CERES-Maize). This indicated that grain yield was not well-simulated accurately for the cultivars. The evaluation statistics for the biomass yield, grain yield, grain unit wt. and grain number $m^{-2}$ are shown in Tables 8 and 9. The overall RMSE for grain (RMSE = 0.84 t ha$^{-1}$, dstat = 0.92, $R^2$ = 0.72) and biomass (RMSE = 2.87 t ha$^{-1}$, d-stat = 0.89, $R^2$ = 0.86) yield in the DSSAT-CERES-Maize model were 11.28% and 64.84% and considered as good and poor, respectively. The simulation of grain (RMSE = 1.38 t ha$^{-1}$, d-stat = 0.57, $R^2$ = 0.26) and biomass (RMSE = 3.19 t ha$^{-1}$, d-stat = 0.90, $R^2$ = 0.85) yield in APSIM-Maize was good and poor, with RMSEn of 18.42% and 71.64%, respectively. However, the simulated biomass in APSIM-Maize was acceptable, with a d-stat $\geq$ 0.70. In contrast, the grain and biomass yield evaluations (DSSAT-CERES-Maize model) for all three cultivars had a d-stat $\geq$ 0.70. The d-stat $\geq$ 0.70 has been reported as being acceptable in crop model evaluations [41,60]. The RMSE for a pooled biomass yield in the APSIM-Maize and DSSAT-CERES-Maize models were 3.19 t ha$^{-1}$ and 2.87 t ha$^{-1}$, respectively. The cultivar RMSE for the grain and biomass (Tables 8 and 9) yield in DSSAT-CERES-Maize ranged from 0.68 to 1.11 t ha$^{-1}$ and 2.65 to 3.03 t ha$^{-1}$, respectively. The evaluation of the DSSAT-CERES-Maize model showed that the model was able to simulate the grain yield for the three cultivars accurately, as shown in Tables 8 and 9. The modeling of the grain yield for ZMS 606 and PHB 30G19 and PHB 30B50 were considered excellent and good, respectively. The modeling of the biomass yield was considered good (PHB 30B50, RMSEn (10–20%)) and acceptable (ZMS 606 and PHB 30G19, RMSEn (20–30%)).

**Table 8.** The APSIM-Maize and DSSAT-CERES-Maize model calibration statistics for the biomass yield for the maize cultivars under rainfed conditions.

| Cultivar | APSIM-Maize | | | | | DSSAT-CERES-Maize | | | | |
|---|---|---|---|---|---|---|---|---|---|---|
| | RMSEn | d-stat | RMSE | ME | $R^2$ | RMSEn | d-stat | RMSE | ME | $R^2$ |
| ZMS 606 | 53.31 | 0.25 | 5.21 | 5.09 | 0.12 | 68.08 | 0.89 | 2.86 | 2.22 | 0.87 |
| PHB 30G19 | 43.30 | 0.30 | 4.56 | 4.41 | 0.19 | 63.74 | 0.89 | 3.03 | 2.40 | 0.89 |
| PHB 30B50 | 43.16 | 0.43 | 4.60 | 4.18 | 0.15 | 55.31 | 0.92 | 2.65 | 1.90 | 0.87 |

### 3.2.3. Grain Unit wt. and Grain Number per Square Meter

The simulation of the grain unit wt. and grain number $m^{-2}$ in the DSSAT-CERES-Maize model was acceptable, with RMSE of 0.07 g and 379.81 grains $m^{-2}$, respectively, as shown in Tables 10 and 11. The underprediction of the grain number $m^{-2}$ by the DSSAT-CERES-Maize model was also reported by [2,61]. In APSIM-Maize, the RMSE for the grain unit wt. and grain number $m^{-2}$ were 0.05 g and 276.13 grains $m^{-2}$, respectively. The simulation of the grain unit wt. and grain number $m^{-2}$ in APSIM-Maize was acceptable, with a d-stat $\geq$ 0.70 and RMSEn of 10–20%. The sowing date analysis showed that delaying the sowing date from 12 December 2016 to 26 December 2016 and 9 January 2017 caused a decrease in the average yield for all cultivars. Other researchers reported that APSIM performs better in simulating the soil water, grain yield and nitrogen dynamics in different agricultural production systems in Australia [52]. In this study, DSSAT-CERES-Maize performed better in the simulation grain yield, grain unit wt. and grain number $m^{-2}$. The DSSAT-CERES-Maize model explained 63%, 86% and 72% of the variation in the observed

grain unit wt., biomass and grain yield, respectively. In APSIM-Maize, 85% of the biomass yield was explained by the model.

**Table 9.** The APSIM-Maize and DSSAT-CERES-Maize model calibration statistics for the grain yield for the maize cultivars under rainfed conditions.

| | APSIM-Maize | | | | | DSSAT-CERES-Maize | | | | |
|---|---|---|---|---|---|---|---|---|---|---|
| Cultivar | RMSEn | d-stat | RMSE | ME | $R^2$ | RMSEn | d-stat | RMSE | ME | $R^2$ |
| ZMS 606 | 17.41 | 0.66 | 1.28 | 1.03 | 0.60 | 9.16 | 0.91 | 0.67 | 0.13 | 0.84 |
| PHB 30G19 | 17.34 | 0.66 | 1.27 | 1.00 | 0.58 | 9.27 | 0.93 | 0.68 | −0.04 | 0.76 |
| PHB 30B50 | 20.13 | 0.44 | 1.56 | −0.13 | 0.62 | 14.31 | 0.91 | 1.11 | 0.08 | 0.69 |

**Table 10.** The APSIM-Maize and DSSAT-CERES-Maize model calibration statistics for the grain unit wt. for the maize cultivars under rainfed conditions.

| | APSIM-Maize | | | | | DSSAT-CERES-Maize | | | | |
|---|---|---|---|---|---|---|---|---|---|---|
| Cultivar | RMSEn | d-stat | RMSE | ME | $R^2$ | RMSEn | d-stat | RMSE | ME | $R^2$ |
| ZMS 606 | 15.05 | 0.61 | 0.05 | 0.02 | 0.19 | 19.25 | 0.55 | 0.06 | −0.02 | 0.57 |
| PHB 30G19 | 12.39 | 0.68 | 0.04 | 0.00 | 0.22 | 22.10 | 0.52 | 0.08 | −0.03 | 0.33 |
| PHB 30B50 | 15.02 | 0.72 | 0.06 | 0.00 | 0.51 | 14.28 | 0.91 | 0.06 | −0.02 | 0.79 |

**Table 11.** The APSIM-Maize and DSSAT-CERES-Maize model calibration statistics for the grain number m$^{-2}$ for the maize cultivars under rainfed conditions.

| | APSIM-Maize | | | | | DSSAT-CERES-Maize | | | | |
|---|---|---|---|---|---|---|---|---|---|---|
| Cultivar | RMSEn | d-stat | RMSE | ME | $R^2$ | RMSEn | d-stat | RMSE | ME | $R^2$ |
| ZMS 606 | 10.21 | 0.42 | 230.55 | 82.89 | 0.00 | 14.66 | 0.35 | 330.92 | 176.89 | 0.00 |
| PHB 30G19 | 12.81 | 0.37 | 283.20 | 188.56 | 0.00 | 16.41 | 0.30 | 362.67 | 194.00 | 0.00 |
| PHB 30B50 | 17.52 | 0.22 | 308.84 | 135.44 | 0.32 | 24.84 | 0.30 | 437.91 | 304.78 | 0.31 |

### 3.2.4. Leaf Area Index

The RMSE for the mLAI in the APSIM-Maize and DSSAT-CERES-Maize models were 0.69 and 2.04 m$^2$ m$^{-2}$, respectively. The simulations of the mLAI in APSIM-Maize and DSSAT-CERES-Maize models were good and poor, respectively. The maximum LAI in the models was underpredicted at all the treatment levels, as shown in Table 12. Similar results in the DSSAT-CERES-Maize model were reported [2]. The APSIM-Maize and DSSAT-CERES-Maize models simulate LAI, but in this situation, the functions performed poorly. The model failed to accurately simulate LAI for the three maize cultivars. LAI provides an index of the maize plant growth and is an important input into the APSIM-Maize and DSSAT-CERES-Maize models [36]. Accurate field or laboratory measurements of LAI are required to evaluate crop models properly. The simulated mLAI ranged from 3.02 to 3.35 m$^2$ m$^{-2}$ (APSIM-Maize) and 1.44 to 2.18 m$^2$ m$^{-2}$ (DSSAT-CERES-Maize), respectively. Typical LAI values of maize cultivars in dry land areas under rainfed production are 2.50–2.90 m$^2$ m$^{-2}$ [74] and agree with the results from the APSIM-Maize model.

### 3.2.5. Simulation of Root Soil Water Content in the Soil Layers

The statistics for pooled root soil water content under irrigated conditions are shown in Table 13 for the APSIM-Maize and DSSAT-CERES-Maize models. The predictive ability of the DSSAT-CERES-Maize model was excellent (RMSEn >10%, soil layer: 4), good (RMSEn 10–20%, soil layers: 2 to 3 and 5–7) and acceptable (RMSEn 20–30%, soil layers: 1 and 8). In

the APSIM-Maize model, the predictive ability of the model was good (RMSEn 10–20%, soil layers: 1 and 6–8) and excellent (RMSEn >10%, soil layers 2–5). The modeling of the root soil water content may have been affected by unmeasured soil parameters, such as management scenarios (weeds, disease and pest control); LL; DUL and SAT. The APSIM soilwat module requires the determination of LL and DUL under field conditions [41]. The root soil water is important and should be modeled accurately, as it affects the crop growth and prediction of the grain and biomass yield [75]. Changes in the available root soil water content lead to variations in the biomass and grain yield for maize cultivars. Maize crops can only use root soil water that is present within the reach of their root system, and this could significantly influence the total available root soil water.

**Table 12.** The APSIM-Maize and DSSAT-CERES-Maize model calibration statistics for mLAI for the maize cultivars under rainfed conditions.

| | APSIM-Maize | | | | | DSSAT-CERES-Maize | | | | |
|---|---|---|---|---|---|---|---|---|---|---|
| Cultivar | RMSEn | d-stat | RMSE | ME | $R^2$ | RMSEn | d-stat | RMSE | ME | $R^2$ |
| ZMS 606 | 7.27 | 0.48 | 0.24 | −0.17 | 0.10 | 51.54 | 0.14 | 1.73 | −1.73 | 0.15 |
| PHB 30G19 | 18.15 | 0.40 | 0.69 | −0.62 | 0.03 | 47.83 | 0.20 | 1.83 | −1.84 | 0.06 |
| PHB 30B50 | 24.30 | 0.42 | 0.94 | −0.70 | 0.06 | 62.91 | 0.29 | 2.42 | −2.37 | 0.60 |

**Table 13.** The APSIM-Maize and DSSAT-CERES-Maize model calibration statistics for the root soil water content under rainfed conditions.

| | APSIM-Maize | | DSSAT-CERES-Maize | |
|---|---|---|---|---|
| Soil Layer (cm$^3$ cm$^{-3}$) | RMSEn | RMSE | RMSEn | RMSE |
| soil layer 1 | 16.08 | 0.04 | 28.95 | 0.09 |
| soil layer 2 | 8.37 | 0.03 | 19.53 | 0.07 |
| soil layer 3 | 7.69 | 0.03 | 10.25 | 0.04 |
| soil layer 4 | 7.34 | 0.03 | 9.79 | 0.04 |
| soil layer 5 | 9.56 | 0.04 | 11.95 | 0.05 |
| soil layer 6 | 11.74 | 0.05 | 11.74 | 0.05 |
| soil layer 7 | 11.69 | 0.05 | 11.69 | 0.05 |
| soil layer 8 | 14.12 | 0.06 | 21.18 | 0.09 |

The analysis of SDs showed that the amount of root soil water available for plant growth and rainfall decreased during each maize growing period, and this affected the biomass and grain yield. Other researchers reported that root soil water profile layers of 5–15, 15–30 and 30–45-cm thickness are important for simulating the correct root soil water balance and plant water uptake [76]. Water stress in maize increased as the rainfall amount decreased during each SD duration, and this contributed to the grain yield reduction, similar to other findings [77].

*3.3. APSIM-Maize and DSSAT-CERES-Maize Model Validation*

3.3.1. Model Validation

Phenology (Anthesis and Maturity Days after Planting)

The observed and simulated phenological stages, grain unit wt., grain number m$^{-2}$, grain and biomass yield. A number of validation statistics have been used to evaluate the models in examining their predictive ability, usefulness and robustness, as shown in Tables 14 and 15. In the APSIM-Maize model, the mLAI, DAP to anthesis and maturity ranged from 3.26 to 3.37 m$^2$ m$^{-2}$, 87 to 88 days and 135 to 137 days under irrigated conditions, respectively. The DAP to anthesis and maturity were good (RMSEn = 10–20%), with RMSE being 16.69 and 17.36 days, respectively as reported by [59]. The mLAI, DAP to anthesis and maturity of the maize cultivars in the DSSAT-CERES-Maize model ranged

from 1.61 to 2.11 m$^2$ m$^{-2}$, 80 to 86 days and 130 to 137 days, respectively. The RMSE for DAP to anthesis (RMSEn = 19.94%) and maturity (RMSEn = 12.77%) were 20.67 and 19.54 days. The vegetative and reproductive phases of the maize cultivars were delayed due to lowering of the temperature under irrigated conditions. Additionally, lowering temperatures during the growing season reduced the plant height through a decreased internode length, and similar results were reported [78].

**Table 14.** Results of the validation statistics for the APSIM-Maize model under irrigated conditions.

|  | Anthesis | Maturity | Grain Yield | Biomass Yield | Grain Size | Grain No m$^{-2}$ | mLAI |
|---|---|---|---|---|---|---|---|
| NRMSE | 16.00 | 12.77 | 20.18 | 73.98 | 6.25 | 18.51 | 16.75 |
| RMSE | 16.69 | 19.54 | 1.34 | 6.87 | 0.02 | 456.21 | 0.57 |
| MAE | 20.67 | 17.33 | 1.01 | 6.77 | 0.01 | 354.44 | 0.50 |
| CRM | 0.20 | 0.11 | −0.11 | −0.73 | −0.04 | −0.09 | 0.01 |
| Pearson | 0.63 | NA | −0.01 | 0.20 | 0.89 | 0.32 | −0.33 |
| d-stat | 0.04 | NA | 0.48 | 0.23 | 0.75 | 0.59 | 0.11 |

**Table 15.** Results of the validation statistics for the CERES-Maize model under irrigated conditions.

|  | Anthesis | Maturity | Grain Yield | Biomass Yield | Grain Size | Grain No m$^{-2}$ | mLAI |
|---|---|---|---|---|---|---|---|
| NRMSE | 19.94 | 11.34 | 45.08 | 71.64 | 21.14 | 26.91 | 49.56 |
| RMSE | 20.80 | 17.36 | 2.99 | 6.65 | 0.06 | 663.29 | 1.67 |
| MAE | 20.67 | 19.33 | 2.79 | 6.51 | 0.05 | 568.89 | 1.59 |
| CRM | 0.20 | 0.13 | −0.42 | −0.70 | −0.18 | −0.22 | 0.47 |
| Pearson | 0.63 | NA | 0.21 | 0.27 | 0.85 | 0.32 | 0.38 |
| d-stat | 0.04 | NA | 0.35 | 0.24 | 0.55 | 0.52 | 0.36 |

The APSIM-Maize and DSSAT-CERES-Maize models simulate DAP to anthesis and maturity, but in this situation, the function performed poorly. The phenological stages (anthesis and maturity) took longer due to lower solar radiation and cooler air temperature during the field experiment. Plant stress during its growth period from silking to maturity affect the grain yield and yield components [28]. Adverse conditions such as lower temperature, soil water content and nitrogen deficiency delay plant growth and slows silk development and maturity [70,71].

Crop simulation models require accurate field observations for the timing of vegetative (V6) and reproductive stages (silking, dough and physiological maturity stages). This is important in crop simulation, so that partitioning of dry matter accumulation and growth duration is determined properly. The staging of phenological stages is vital in field experiments, since DAP to anthesis and physiological maturity are important determinants of productivity [43].

3.3.2. Biomass, Grain Yield and Leaf Area Index

The calibrated model was validated against the end-of-season mLAI, grain unit wt., grain number m$^{-2}$, biomass and grain yield. The validation of the APSIM-Maize model was the grain unit wt. (RMSE = 0.02 g, RMSEn = 6.25% and d-stat = 0.75); grain number m$^{-2}$ (RMSE = 456.21 grain m$^{-2}$, RMSEn = 18.51% and d-stat = 0.59); biomass (RMSE = 6.87 t ha$^{-1}$ and d-stat = 0.23) and grain yield (RMSE = 1.34 t ha$^{-1}$, RMSEn = 20.18% and d-stat = 0.48). The simulation of the grain yield, grain unit wt. grain number m$^{-2}$ and mLAI showed the robustness of the APSIM-Maize model considered acceptable in crop modeling [59].

Using the DSSAT-CERES-Maize model, the RMSE for grain unit wt. (RMSEn = 21.14%, R$^2$ = 0.72 and d-stat = 0.55); grain number m$^{-2}$ (RMSEn = 26.91% and d-stat = 0.52); grain (RMSEn = 45.08% and d-stat = 0.35) and biomass (RMSEn = 71.64% and d-stat = 0.24) yield

were 0.06 g, 663.29 grains $m^{-2}$, 2.99 t $ha^{-1}$ and 6.65 t $ha^{-1}$, respectively. The simulation of the grain unit wt. and grain number $m^{-2}$ were evaluated to be good. However, the simulation of the grain (RMSEn = 45.08%) and biomass (RMSEn = 71.64%) yield were considered to be poor in crop modeling, as reported by [59]. The RMSE for mLAI in APSIM-Maize (RMSEn = 16.75%) and DSSAT-CERES-Maize (RMSEn = 49.56%) were 0.57 $m^2$ $m^{-2}$ and 1.67 $m^2$ $m^{-2}$ and considered acceptable and poor in the crop model evaluation, respectively. The accuracy of the simulated outcomes of the crop models using new cultivars depend on rigorous calibrations to minimize uncertainties.

## 4. Conclusions

The results showed that the maize grain yield and yield components are affected by varying sowing dates and cultivar and N application rates. Farmers could enhance maize yields by manipulating the sowing date, cultivar selection and application of nitrogen fertilizer in agriculture production systems. The treatment effect of the sowing date and cultivar are significant on the biomass yield, 100-grain weight, seed number $m^{-2}$, cob length and cob width. The grain yield, aboveground biomass, seed number $m^{-2}$ and 100-grain wt. were reduced with a delay in the sowing date of either 7 or 14 days. The reduction in grain yield from SD1-SD2 (1.91 t $ha^{-1}$), SD1-SD3 (2.90 t $ha^{-1}$) and SD2-SD3 (0.99 t $ha^{-1}$) were 21.04, 31.83 and 13.83%, respectively under the rainfed experiment.

The APSIM-Maize and DSSAT-CERES-Maize model calibration parameters are quite demanding. The models require complete data on the phenology, biomass, grain yield, grain number $m^{-2}$, grain unit wt., rate of grain filling in milligram $day^{-1}$ and leaf area index. The two crop simulation models provided acceptable simulation of DAP to anthesis, physiological maturity, biomass and grain yield. The models appear to be similar in their predictive ability in providing acceptable simulation. The crop models could be used as decision support tools by agricultural research stations, policymakers, farmers and planners for forecasting crop yield. Crop growth and yield can be simulated successfully using crop growth models before establishing costly and labor-intensive field experiments. Farmers and researchers can use crop models in investigating the effects of different management scenarios (plant density, planting time, fertilizer applications, irrigation regime and scheduling) and crop yield forecasting using the calibrated and validated maize cultivars.

The APSIM-Maize and DSSAT-CERES-Maize model simulates DAP to anthesis and maturity, but in this study, the function performed poorly. The plant growth duration took longer than expected due to lower solar radiation and cooler air temperature under irrigated condition. The key areas for APSIM-Maize and DSSAT-CERES-Maize models improvement includes the simulation of phenological stage duration and LAI prediction during winter and summer in the subtropical environment.

**Author Contributions:** C.B.C. and E.P. developed the study, designed the field experiments and selected the study area. C.B.C. monitored the field experiments, collected and prepared, and analyzed the data. C.B.C. and E.P. prepared the articles. C.B.C. created and prepared the tables and figures. V.R.N.C. proofread the articles and also checked the analysis and conclusion. All authors have read and agreed to the published version of the manuscript.

**Funding:** This research received no external funding.

**Data Availability Statement:** Data can be requested from the corresponding author.

**Acknowledgments:** The researchers wish to thank Zambia Agricultural Research Institute (ZARI) for providing research facilities for both irrigated and rainfed experiments.

**Conflicts of Interest:** The authors declare no conflict of interest.

## Appendix A

**Table A1.** Summary of the observed data collected from the rainfed experiment site. Reproduced with permission from [33].

| | SD1 | | | | | | | | | SD2 | | | | | | | | | SD3 | | | | | | | | |
|---|---|---|---|---|---|---|---|---|---|---|---|---|---|---|---|---|---|---|---|---|---|---|---|---|---|---|---|
| Cultivars | ZMS 606 | | | 30G19 | | | PHB 30B50 | | | ZMS 606 | | | PHB 30G19 | | | PHB 30B50 | | | ZMS 606 | | | PHB 30G19 | | | PHB 30B50 | | |
| N rate | 1 | 2 | 3 | 1 | 2 | 3 | 1 | 2 | 3 | 1 | 2 | 3 | 1 | 2 | 3 | 1 | 2 | 3 | 1 | 2 | 3 | 1 | 2 | 3 | 1 | 2 | 3 |
| Land preparation | | | | | | | | | | | | | 29 November 2016 | | | | | | | | | | | | | | |
| Basal dressing/planting | | | | | 12 December 2016 | | | | | | | | 26 December 2016 | | | | | | | | | | 9 January 2017 | | | | |
| Top dressing | | | | | 30 January 2017 | | | | | | | | 17 February 2017 | | | | | | | | | | 3 March 2017 | | | | |
| Herbicides | | | | | | | | | | | | | 14 December 2016 | | | | | | | | | | | | | | |
| Herbicides | | | | | | | | | | | | | 23 December 2016 & 18 January 2017 | | | | | | | | | | | | | | |
| Weeding | | | | | | | | | | | | | 17 January 2017 | | | | | | | | | | | | | | |
| Pesticides | | | | | | | | | | | 29 December 2016 | | | | | | | | | | | | | | | | |
| | | | | | | | | | | Phenological stages | | | | | | | | | | | | | | | | |
| Emergence | 19 December 2016 | | | 19 December 2016 | | | 89 December 2016 | | | 4 January 2017 | | | 4 January 2017 | | | 3 January 2017 | | | 17 January 2017 | | | 16 January 2017 | | | 17 January 2017 | | |
| V6 | 6 January 2017 | | | 6 January 2017 | | | 6 January 2017 | | | 20 January 2017 | | | 20 January 2017 | | | 19 January 2017 | | | 6 February 2017 | | | 6 February 2017 | | | 5 February 2017 | | |
| R1 | 15 February 2017 | | | 15 February 2017 | | | 13 February 2017 | | | 4 March 2017 | | | 2 March 2017 | | | 4 March 2017 | | | 19 March 2017 | | | 19 March 2017 | | | 17 March 2017 | | |
| R4 | 14 March 2017 | | | 19 March 2017 | | | 12 March 2017 | | | 28 March 2017 | | | 28 March 2017 | | | 26 March 2017 | | | 12 April 2017 | | | 12 April 2017 | | | 10 April 2017 | | |
| R6 | 14 April 2017 | | | 15 April 2017 | | | 13 April 2017 | | | 28 April 2017 | | | 27 April 2017 | | | 27 April 2017 | | | 18 May 2017 | | | 18 May 2017 | | | 19 May 2017 | | |
| | | | | | | | | | | Biomass sampling | | | | | | | | | | | | | | | | |
| V6 | 6 January 2017 | | | 6 January 2017 | | | 6 January 2017 | | | 20 January 2017 | | | 20 January 2017 | | | 20 January 2017 | | | 6 February 2017 | | | 6 February 2017 | | | 6 February 2017 | | |
| R1 | 15 February 2017 | | | 15 February 2017 | | | 13 February 2017 | | | 4 March 2017 | | | 4 March 2017 | | | 2 March 2017 | | | 21 March 2017 | | | 21 March 2017 | | | 21 March 2017 | | |
| R4 | 16 March 2017 | | | 16 March 2017 | | | 16 March 2017 | | | 30 March 2017 | | | 30 March 2017 | | | 28 March 2017 | | | 13 April 2017 | | | 13 April 2017 | | | 13 April 2017 | | |
| Final harvest | 3 May 2017 | | | | | | | | | 15 May 2017 | | | | | | | | | 1 January 2017 | | | | | | | | |

1 = N1; 2 = N2; 3 = N3; Pesticide = Monocrotophos, Fustac; Herbicide = Nicosulfuron; Termites: Terminator (Imidacloprid 30.5% SC) 350 g of Imidacloprid/liter.

**Table A2.** Comparison between the observed and simulated phenology, mLAI, biomass, grain and grain size in the CERES-Maize model under the rainfed condition. Reproduced with permission from [61].

| | Anthesis | | | Maturity | | | mLAI | | | Biomass | | | Grain | | | Unit wt | | | Grain no | | |
|---|---|---|---|---|---|---|---|---|---|---|---|---|---|---|---|---|---|---|---|---|---|
| | | | | | | | | | Sowing Date 1 | | | | | | | | | | | | |
| Trt | Obs | Sim | %PD | Obs | Sim | %PD | Obs | Sim | %PD | Obs | Sim | %PD | Obs | Sim | %PD | Obs | Sim | %PD | Obs | Sim | %PD |
| V1N1 | 65 | 64 | −1.54 | 123 | 123 | 0.00 | 3.70 | 1.54 | −58.38 | 11.40 | 13.70 | 20.18 | 9.20 | 8.80 | −4.35 | 0.37 | 0.38 | 2.15 | 2401 | 2322 | 6.91 |
| V1N2 | 65 | 64 | −1.54 | 123 | 123 | 0.00 | 3.29 | 1.55 | −52.89 | 10.90 | 13.70 | 25.69 | 8.70 | 8.70 | 0.00 | 0.38 | 0.38 | 0.53 | 2276 | 2328 | 2.28 |
| V1N3 | 65 | 64 | −1.54 | 123 | 123 | 0.00 | 3.15 | 1.56 | −50.48 | 9.80 | 13.80 | 40.82 | 7.90 | 8.80 | 11.39 | 0.37 | 0.37 | 0.00 | 2128 | 2341 | 10.01 |
| V2N1 | 65 | 65 | 0.00 | 124 | 124 | 0.00 | 3.79 | 1.96 | −48.28 | 11.20 | 14.80 | 32.14 | 8.40 | 8.80 | 4.76 | 0.37 | 0.38 | 2.70 | 2251 | 2297 | 2.04 |
| V2N2 | 65 | 65 | 0.00 | 124 | 124 | 0.00 | 3.68 | 1.97 | −46.47 | 10.50 | 14.70 | 40.00 | 7.90 | 8.80 | 11.39 | 0.41 | 0.38 | −7.32 | 1922 | 2299 | 19.61 |
| V2N3 | 65 | 65 | 0.00 | 124 | 124 | 0.00 | 4.45 | 1.98 | −55.51 | 13.10 | 14.80 | 12.98 | 9.70 | 8.80 | −9.28 | 0.38 | 0.38 | 0.00 | 2519 | 2305 | −8.50 |

**Table A2.** *Cont.*

| Trt | Anthesis | | | Maturity | | | mLAI | | | Biomass | | | Grain | | | Unit wt | | | Grain no | | |
|---|---|---|---|---|---|---|---|---|---|---|---|---|---|---|---|---|---|---|---|---|---|
| V3N1 | 63 | 64 | 1.59 | 122 | 122 | 0.00 | 4.22 | 1.54 | −63.51 | 12.50 | 14.40 | 15.20 | 9.50 | 10.10 | 6.32 | 0.51 | 0.51 | 0.00 | 1867 | 1996 | 6.91 |
| V3N2 | 63 | 64 | 1.59 | 122 | 122 | 0.00 | 4.56 | 1.55 | −66.01 | 12.70 | 14.50 | 14.17 | 9.60 | 10.20 | 6.25 | 0.50 | 0.51 | 2.00 | 1952 | 2001 | 2.51 |
| V3N3 | 63 | 64 | 1.59 | 122 | 122 | 0.00 | 5.00 | 1.56 | −68.80 | 14.20 | 14.50 | 2.11 | 10.80 | 10.20 | −5.56 | 0.52 | 0.51 | −1.92 | 2063 | 2012 | −2.47 |
| Sowing Date 2 | | | | | | | | | | | | | | | | | | | | | |
| V1N1 | 67 | 64 | −4.48 | 123 | 127 | 3.25 | 3.57 | 1.56 | −56.30 | 9.10 | 12.00 | 31.87 | 6.90 | 7.60 | 10.14 | 0.32 | 0.36 | 12.50 | 2113 | 2195 | 3.88 |
| V1N2 | 67 | 64 | −4.48 | 123 | 127 | 3.25 | 3.45 | 1.56 | −54.78 | 10.40 | 11.90 | 14.42 | 8.20 | 7.50 | −8.54 | 0.34 | 0.34 | 0.00 | 2421 | 2194 | −9.38 |
| V1N3 | 67 | 64 | −4.48 | 123 | 127 | 3.25 | 3.26 | 1.60 | −50.92 | 8.60 | 12.00 | 39.53 | 6.50 | 7.60 | 16.92 | 0.30 | 0.34 | 13.33 | 2190 | 2218 | 1.28 |
| V2N1 | 66 | 65 | −1.52 | 122 | 128 | 4.92 | 3.75 | 1.91 | −49.07 | 10.20 | 12.80 | 25.49 | 7.30 | 7.60 | 4.11 | 0.34 | 0.36 | 5.88 | 2146 | 2146 | 0.00 |
| V2N2 | 66 | 65 | −1.52 | 122 | 128 | 4.92 | 3.58 | 1.93 | −46.09 | 9.30 | 12.70 | 36.56 | 6.50 | 7.50 | 15.38 | 0.30 | 0.35 | 16.67 | 2125 | 2154 | 1.36 |
| V2N3 | 66 | 65 | −1.52 | 122 | 128 | 4.92 | 4.08 | 1.94 | −52.45 | 11.20 | 12.90 | 15.18 | 7.90 | 7.70 | −2.53 | 0.34 | 0.36 | 5.88 | 2311 | 2158 | −6.62 |
| V3N1 | 68 | 62 | −8.82 | 122 | 123 | 0.82 | 3.64 | 1.47 | −59.62 | 10.50 | 12.10 | 15.24 | 7.90 | 7.90 | 0.00 | 0.43 | 0.42 | −2.33 | 1861 | 1908 | 2.53 |
| V3N2 | 68 | 62 | −8.82 | 122 | 123 | 0.82 | 3.20 | 1.47 | −54.06 | 7.30 | 12.00 | 64.38 | 5.30 | 7.90 | 49.06 | 0.35 | 0.41 | 17.14 | 1497 | 1907 | 27.39 |
| V3N3 | 68 | 62 | −8.82 | 122 | 123 | 0.82 | 3.18 | 1.50 | −52.83 | 11.00 | 12.30 | 11.82 | 8.00 | 8.10 | 1.25 | 0.37 | 0.42 | 13.51 | 1908 | 1929 | 1.10 |
| Sowing Date 3 | | | | | | | | | | | | | | | | | | | | | |
| V1N1 | 66 | 65 | −1.52 | 129 | 130 | 0.78 | 3.18 | 1.60 | −49.69 | 10.10 | 10.80 | 6.93 | 6.00 | 5.80 | −3.33 | 0.32 | 0.21 | −34.38 | 2300 | 2793 | 21.43 |
| V1N2 | 66 | 65 | −1.52 | 129 | 130 | 0.78 | 3.34 | 1.59 | −52.40 | 8.90 | 10.70 | 20.22 | 6.50 | 5.80 | −10.77 | 0.31 | 0.21 | −32.26 | 2425 | 2755 | 13.61 |
| V1N3 | 66 | 65 | −1.52 | 129 | 130 | 0.78 | 3.34 | 1.59 | −52.40 | 8.70 | 10.60 | 21.84 | 6.10 | 5.60 | −8.20 | 0.32 | 0.20 | −37.50 | 2068 | 2789 | 34.86 |
| V2N1 | 69 | 68 | −1.45 | 129 | 134 | 3.88 | 3.61 | 2.18 | −39.61 | 9.50 | 11.70 | 23.16 | 6.10 | 5.30 | −13.11 | 0.31 | 0.19 | −38.71 | 2043 | 2768 | 35.49 |
| V2N2 | 69 | 68 | −1.45 | 129 | 134 | 3.88 | 4.02 | 2.18 | −45.77 | 9.60 | 11.80 | 22.92 | 6.10 | 5.50 | −9.84 | 0.33 | 0.20 | −39.39 | 2243 | 2765 | 23.27 |
| V2N3 | 69 | 68 | −1.45 | 129 | 134 | 3.88 | 3.40 | 2.17 | −36.18 | 10.10 | 11.90 | 17.82 | 6.10 | 5.60 | −8.20 | 0.33 | 0.20 | −39.55 | 2331 | 2745 | 17.76 |
| V3N1 | 67 | 62 | −7.46 | 130 | 127 | −2.31 | 3.55 | 1.45 | −59.15 | 8.40 | 10.30 | 22.62 | 5.60 | 5.50 | −1.79 | 0.35 | 0.24 | −31.43 | 1568 | 2291 | 46.11 |
| V3N2 | 67 | 62 | −7.46 | 130 | 127 | −2.31 | 3.59 | 1.44 | −59.89 | 10.00 | 10.10 | 1.00 | 6.90 | 5.40 | −21.74 | 0.33 | 0.24 | −27.27 | 1628 | 2266 | 39.19 |
| V3N3 | 67 | 62 | −7.46 | 130 | 127 | −2.31 | 3.72 | 1.44 | −61.29 | 9.30 | 10.10 | 8.60 | 6.30 | 5.30 | −15.87 | 0.29 | 0.23 | −20.69 | 1521 | 2298 | 51.08 |

DAP = Days after planting (anthesis, physiological maturity); mLAI ($m^2\ m^{-2}$) = maximum leaf area index, biomass (t ha$^{-1}$), unity weight (g); Negative deviations indicate under prediction while positive deviations indicate over prediction; %PD = percentage prediction deviation, Sim = Simulated; Obs = observed; DAP = Days after planting; mLAI = maximum leaf area index; V1 = ZMS 606; V2 = PHB 30G19; V3 = PHB 30B50; Trt = Treatment.

**Table A3.** Comparison between the observed and simulated phenology, mLAI, biomass, grain and grain size in the APSIM-Maize model under the rainfed condition. Reproduced with permission from [61].

| | Anthesis | | | Maturity | | | mLAI | | | Biomass | | | Grain | | | Unit wt | | | Grain no | | |
|---|---|---|---|---|---|---|---|---|---|---|---|---|---|---|---|---|---|---|---|---|---|
| | | | | | | | | | | Sowing Date 1 | | | | | | | | | | | |
| Trt | Obs | Sim | %PD | Obs | Sim | %PD | Obs | Sim | %PD | Obs | Sim | %PD | Obs | Sim | %PD | Obs | Sim | %PD | Obs | Sim | %PD |
| V1N1 | 65 | 65 | 0.00 | 123 | 124 | 0.81 | 3.70 | 3.18 | −14.05 | 11.40 | 15.40 | 35.09 | 9.20 | 8.90 | −3.26 | 0.37 | 0.39 | 4.84 | 2401 | 2261 | −0.86 |
| V1N2 | 65 | 65 | 0.00 | 123 | 124 | 0.81 | 3.29 | 3.19 | −3.04 | 10.90 | 15.40 | 41.28 | 8.70 | 8.90 | 2.30 | 0.38 | 0.39 | 3.17 | 2276 | 2262 | −0.62 |
| V1N3 | 65 | 65 | 0.00 | 123 | 124 | 0.81 | 3.15 | 3.19 | 1.27 | 9.80 | 15.40 | 57.14 | 7.90 | 8.90 | 12.66 | 0.37 | 0.39 | 5.41 | 2128 | 2262 | 6.30 |
| V2N1 | 65 | 65 | 0.00 | 124 | 124 | 0.00 | 3.79 | 3.19 | −15.83 | 11.20 | 15.50 | 38.39 | 8.40 | 8.90 | 5.95 | 0.37 | 0.38 | 2.70 | 2251 | 2324 | 3.24 |
| V2N2 | 65 | 65 | 0.00 | 124 | 124 | 0.00 | 3.68 | 3.19 | −13.32 | 10.50 | 15.50 | 47.62 | 7.90 | 8.90 | 12.66 | 0.41 | 0.38 | −7.32 | 1922 | 2324 | 20.92 |
| V2N3 | 65 | 65 | 0.00 | 124 | 124 | 0.00 | 4.45 | 3.19 | −28.31 | 13.10 | 15.50 | 18.32 | 9.70 | 8.90 | −8.25 | 0.38 | 0.38 | 0.00 | 2519 | 2324 | −7.74 |
| V3N1 | 63 | 65 | 3.17 | 122 | 123 | 0.82 | 4.22 | 3.14 | −25.59 | 12.50 | 15.30 | 22.40 | 9.50 | 8.00 | −15.79 | 0.51 | 0.43 | −15.69 | 1867 | 1851 | −0.86 |
| V3N2 | 63 | 65 | 3.17 | 122 | 123 | 0.82 | 4.56 | 3.14 | −31.14 | 12.70 | 15.30 | 20.47 | 9.60 | 8.00 | −16.67 | 0.50 | 0.43 | −14.00 | 1952 | 1852 | −5.12 |
| V3N3 | 63 | 65 | 3.17 | 122 | 123 | 0.82 | 5.00 | 3.14 | −37.20 | 14.20 | 15.30 | 7.75 | 10.80 | 8.00 | −25.93 | 0.52 | 0.43 | −17.31 | 2063 | 1852 | −10.23 |
| | | | | | | | | | | Sowing Date 2 | | | | | | | | | | | |
| V1N1 | 67 | 67 | 0.00 | 123 | 130 | 5.69 | 3.57 | 3.34 | −6.44 | 9.10 | 15.40 | 69.23 | 6.90 | 8.70 | 26.09 | 0.32 | 0.40 | 25.00 | 2113 | 2175 | 2.93 |
| V1N2 | 67 | 67 | 0.00 | 123 | 130 | 5.69 | 3.45 | 3.34 | −3.19 | 10.40 | 15.80 | 51.92 | 8.20 | 8.70 | 6.10 | 0.34 | 0.40 | 17.65 | 2421 | 2175 | −10.16 |
| V1N3 | 67 | 67 | 0.00 | 123 | 130 | 5.69 | 3.26 | 3.34 | 2.45 | 8.60 | 15.80 | 83.72 | 6.50 | 8.70 | 33.85 | 0.30 | 0.40 | 33.33 | 2190 | 2175 | −0.68 |
| V2N1 | 66 | 67 | 1.52 | 122 | 128 | 4.92 | 3.75 | 3.34 | −10.93 | 10.20 | 15.80 | 54.90 | 7.30 | 8.80 | 20.55 | 0.34 | 0.38 | 11.76 | 2146 | 2290 | 6.71 |
| V2N2 | 66 | 67 | 1.52 | 122 | 128 | 4.92 | 3.58 | 3.34 | −6.70 | 9.30 | 15.80 | 69.89 | 6.50 | 8.80 | 35.38 | 0.30 | 0.38 | 26.67 | 2125 | 2290 | 7.76 |
| V2N3 | 66 | 67 | 1.52 | 122 | 128 | 4.92 | 4.08 | 3.35 | −17.89 | 11.20 | 15.80 | 41.07 | 7.90 | 8.80 | 11.39 | 0.34 | 0.38 | 11.76 | 2311 | 2290 | −0.91 |
| V3N1 | 68 | 65 | −4.41 | 122 | 125 | 2.46 | 3.64 | 3.29 | −9.62 | 10.50 | 15.60 | 48.57 | 7.90 | 7.70 | −2.53 | 0.43 | 0.43 | 0.00 | 1861 | 1790 | −3.82 |
| V3N2 | 68 | 65 | −4.41 | 122 | 125 | 2.46 | 3.20 | 3.29 | 2.81 | 7.30 | 15.60 | 113.70 | 5.30 | 7.70 | 45.28 | 0.35 | 0.43 | 22.86 | 1497 | 1790 | 19.57 |
| V3N3 | 68 | 65 | −4.41 | 122 | 125 | 2.46 | 3.18 | 3.29 | 3.46 | 11.00 | 15.60 | 41.82 | 8.00 | 7.70 | −3.75 | 0.37 | 0.43 | 16.22 | 1908 | 1790 | −6.18 |
| | | | | | | | | | | Sowing Date 3 | | | | | | | | | | | |
| V1N1 | 66 | 65 | −1.52 | 129 | 130 | 0.78 | 3.18 | 3.05 | −4.09 | 10.10 | 13.50 | 33.66 | 6.00 | 7.50 | 25.00 | 0.32 | 0.29 | −9.38 | 2300 | 2586 | 12.43 |
| V1N2 | 66 | 65 | −1.52 | 129 | 130 | 0.78 | 3.34 | 3.05 | −8.68 | 8.90 | 13.50 | 51.69 | 6.50 | 7.50 | 15.38 | 0.31 | 0.29 | −6.45 | 2425 | 2586 | 6.64 |
| V1N3 | 66 | 65 | −1.52 | 129 | 130 | 0.78 | 3.34 | 3.05 | −8.68 | 8.70 | 13.50 | 55.17 | 6.10 | 7.50 | 22.95 | 0.32 | 0.29 | −9.38 | 2068 | 2586 | 25.05 |
| V2N1 | 69 | 66 | −4.35 | 129 | 129 | 0.00 | 3.61 | 3.05 | −15.51 | 9.50 | 13.50 | 42.11 | 6.10 | 7.30 | 19.67 | 0.31 | 0.28 | −9.68 | 2043 | 2582 | 26.38 |
| V2N2 | 69 | 66 | −4.35 | 129 | 129 | 0.00 | 4.02 | 3.05 | −24.13 | 9.60 | 13.50 | 40.63 | 6.10 | 7.30 | 19.67 | 0.33 | 0.28 | −15.15 | 2243 | 2582 | 15.11 |
| V2N3 | 69 | 66 | −4.35 | 129 | 129 | 0.00 | 3.40 | 3.05 | −10.29 | 10.10 | 13.50 | 33.66 | 6.10 | 7.30 | 19.67 | 0.33 | 0.28 | −15.15 | 2331 | 2582 | 10.77 |
| V3N1 | 67 | 64 | −4.48 | 130 | 128 | −1.54 | 3.55 | 3.02 | −14.93 | 8.40 | 13.60 | 61.90 | 5.60 | 7.20 | 28.57 | 0.35 | 0.35 | 0.00 | 1568 | 2053 | 30.93 |
| V3N2 | 67 | 64 | −4.48 | 130 | 128 | −1.54 | 3.59 | 3.02 | −15.88 | 10.00 | 13.60 | 36.00 | 6.90 | 7.20 | 4.35 | 0.33 | 0.35 | 6.06 | 1628 | 2053 | 26.11 |
| V3N3 | 67 | 64 | −4.48 | 130 | 128 | −1.54 | 3.72 | 3.02 | −18.82 | 9.30 | 13.60 | 46.24 | 6.30 | 7.20 | 14.29 | 0.29 | 0.35 | 20.69 | 1521 | 2053 | 34.98 |

DAP = Days after planting (anthesis, physiological maturity); mLAI (m$^2$ m$^{-2}$) = maximum leaf area index, biomass (t ha$^{-1}$), unity weight (g); Negative deviations indicate under prediction while positive deviations indicate over prediction; %PD = percentage prediction deviation, Sim = Simulated; Obs = observed; DAP = Days after planting; mLAI = maximum leaf area index; V1 = ZMS 606; V2 = PHB 30G19; V3 = PHB 30B50; Trt = Treatment.

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
