# Peer review of "Evaluating APSIM-and-DSSAT-CERES-Maize Models under Rainfed Conditions Using Zambian Rainfed Maize Cultivars"

_nitrogen, doi:10.3390/nitrogen2040027_

Round 1
Reviewer 1 Report
The article contributes to the Nitrogen journal, however, is important to add some comments that I already did to the manuscript body.
In Figure 1 please write in Z axis Temp (°C).
In Figure 2 in the caption write daily precipitation instead of monthly precipitation and Z axis write Temp (°C).
In all manuscript body please write t ha-1, m2, mg day-1 instead of ton ha-1, m-2, milligrams day-1.
In the discussion section please add one or two discussions with other researchers (compare findings)

Author Response
Reviewer comments
Major revisions have been done according to the review comments as explained below.
All the comments and corrections in the PDF files have been incorporated in the MS Office file.
The title has been changed to "Evaluating APSIM-and-DSSAT-CERES_Maize models under rainfed conditions using Zambian rainfed Maize cultivars"
ton ha-1 has been changed to t ha-1 throughout the manuscript
Lines 57-62: this section was deleted as it was redundant
Lines 307-323: this passage was moved to the Materials and Methods section as suggested by the reviewer
Lines 328-348: ANOVA tables have been inserted for rainfed and irrigated condition
Line 374 has been corrected
Figures 1 and 2 have been revised
Calibration figures for DAP to anthesis & maturity, grain yield, biomass yield, LAI, and grain number per square meter have been added to the document
Calibration Figures and validation Tables have been inserted into the manuscript
Line 374 and 427, The number of data points was small, thus R2 was 0. The lines have been deleted
Line 473-483: this section was moved to materials and methods
Line 486: Tables of validation statistics have been provided in the manuscript
Line 530, some more results have been added in the conclusions
More discussion has been added under section 3.3.1
Additional Tables have been added as Tables A2 and A3
Reviewer 2 Report
Title: Evaluating APSIM-and-DSSAT-CERES_Maize models under rainfed conditions using Zambian Maize cultivars: model calibration and validation
Article Type: Research paper
Journal: Nitrogen
General comment: The paper is written well but poorly presented. Authors mentioned in objectives that models were calibrated for sowing dates and nitrogen but there is no results. Also, ANOVA table is missing. In addition, observed meteorological data presented in results. Did they do sme analysis ?. After calibration no evaluation. There is need of several correction to improve the manuscript.
Comment:
1. Title need to modified from “Evaluating APSIM-and-DSSAT-CERES_Maize models under rainfed conditions using Zambian Maize cultivars: model calibration and validation” to “Evaluating APSIM-and-DSSAT-CERES_Maize models under rainfed conditions using Zambian rainfed Maize cultivars”
2. Lines 57-62, these lines are totally redundant. No results reported using RZWQM then why these lines?
3. Lines 307-323, is it output of models if not then shift this to material and methodology section.
4 . Lines 328-348, where is anova table with treatments ?
5. Line 374, Authors need to explain why R2 was 0. Where is the calibration figure ?
6. Line 427, Why R2 is 0 ? calibration figur
7. Line 473, again weather data, is it model results?
8. Line 486, where is the tables of validation results?
9. Line 530, Add some results too in conclusions.
Recommendation: A major revision is required for the manuscript. Calibration graphs is missing and validation tables is missing. Additionally, table also is missing.
Author Response
Reviewer comments
All the comments and corrections in the PDF files have been incorporated in the MS Office file.
The title has been changed to "Evaluating APSIM-and-DSSAT-CERES_Maize models under rainfed conditions using Zambian rainfed Maize cultivars"
ton ha-1 has been changed to t ha-1 throughout the manuscript
Lines 57-62: this section was deleted as it was redundant
Lines 307-323: this passage was moved to the Materials and Methods section as suggested by the reviewer
Lines 328-348: ANOVA tables have been inserted for rainfed and irrigated condition
Line 374 has been corrected
Figures 1 and 2 have been revised
Calibration figures for DAP to anthesis & maturity, grain yield, biomass yield, LAI, and grain number per square meter have been added to the document
Calibration Figures and validation Tables have been inserted into the manuscript
Line 374 and 427, The number of data points was small, thus R2 was 0. The lines have been deleted
Line 473-483: this section was moved to materials and methods
Line 486: Tables of validation statistics have been provided in the manuscript
Line 530, some more results have been added in the conclusions
More discussion has been added under section 3.3.1
Additional Tables have been added as Tables A2 and A3
Round 2
Reviewer 2 Report
This paper improved a lot but still correction is required as the presentation of table and captions are not good.
1) ANOVA results presented in table 3-5 and the variables are not the same. Where id the grain yield for rainfed. all variable for rainfed and irrigated must be same with units which is also missing for irrigated maize.
2) Calibration graph, it is hard to identify which one is model simulated and which one is observed value. Modify it.
3) Tables 6-12, it is hard to understadn which one is rainfed or irrigated. Rewrite the captions.
4) simulation for root soil water content was done for rainfed or irrigated or both. You have not mentioned in text.
Suggestion: Pay attention to presentation of results and the captions. After this correction. Manuscript must be accepted.
Author Response
Response to Reviewer Comments
Point 1) ANOVA results are presented in table 3-5 and the variables are not the same. Where id the grain yield for rainfed. All variables for rainfed and irrigated must be the same with units which are also missing for irrigated maize.
The ANOVA Tables have been corrected and results rewrote to reflect what is presented in the tables. More field observation data was collected for the rainfed than the irrigated field experiments. The rainfed ANOVA has more columns than the irrigated. More discussion has also been added to sections 3.1.1 and 3.1.2.
Point 2) Calibration graph, it is hard to identify which one is model-simulated and which one is observed value. Modify it.
The captions on x-and-y labels for the calibration graphs have been collected.
Point 3) Tables 6-12, it is hard to understand which one is rainfed or irrigated. Rewrite the captions.
The captions for Tables 6-12 has been rewritten to specify that it is either rainfed or irrigated.
Point 4) simulation for root soil water content was done for rainfed or irrigated or both. You have not mentioned in text.
Text has been added to the document specifying which experiment had the soil root water content simulation.
The presentation of results has been addressed. The captions under Results have also been revised and corrected.